# MuSLR: Multimodal Symbolic Logical Reasoning

**Jundong Xu**[1], **Hao Fei**[1],[*] **Yuhui Zhang**[2], **Liangming Pan**[3], **Qijun Huang**[4], **Qian Liu**[5],
**Preslav Nakov**[6], **Min-Yen Kan**[1], **William Yang Wang**[7], **Mong-Li Lee**[1], **Wynne Hsu**[1]

[1] National University of Singapore, [2] Stanford University, [3] Peking University, [4] UniMelb,
[5] University of Auckland, [6] MBZUAI, [7] University of California, Santa Barbara

jundong.xu@u.nus.edu, haofei37@nus.edu.sg, yuhuiz@cs.stanford.edu,
liangmingpan@pku.edu.cn, qijunhuang@student.unimelb.edu.au,
liu.qian@auckland.ac.nz, preslav.nakov@mbzuai.ac.ae, knmnyn@nus.edu.sg,
william@cs.ucsb.edu, dcsleeml@nus.edu.sg, dcshsuw@nus.edu.sg

## Abstract

Multimodal symbolic logical reasoning, which aims to deduce new facts from multimodal input via formal logic, is critical in high-stakes applications such as autonomous driving and medical diagnosis, as its rigorous, deterministic reasoning helps prevent serious consequences. To evaluate such capabilities of current state-of-the-art vision language models (VLMs), we introduce **MuSLR**, the first multimodal symbolic logical reasoning grounded in formal logical rules. We curate a benchmark dataset for MuSLR comprising 1,093 instances across 7 domains, including 35 atomic symbolic logic and 976 logical combinations, with reasoning depths ranging from 2 to 9. We evaluate 7 state-of-the-art VLMs on our benchmark and find that they all struggle with multimodal symbolic reasoning, with the best model, GPT-4.1, achieving only 46.8%. Thus, we propose **LogiCAM**, a modular framework that applies formal logical rules to multimodal inputs, boosting GPT-4.1's Chain-of-Thought performance by 14.13%, and delivering even larger gains on complex logics such as first-order logic. We also conduct a comprehensive error analysis, showing that around 70% of failures stem from logical misalignment between modalities, offering key insights to guide future improvements. All data and code are publicly available at `https://llm-symbol.github.io/MuSLR`.

## 1 Introduction

Recent progress has extensively highlighted the pivotal role of reasoning capabilities in enhancing the generality and robustness of large language models (LLMs) [9, 12, 29–31]. Yet, achieving human-level intelligence demands more than commonsense or heuristic thinking. In particular, *symbolic logical reasoning*, grounded in formal logic such as first-order logic, offers a rigorous, precise, and verifiable paradigm essential for high-stakes scenarios where reasoning errors can have critical consequences. Although previous works have shown that LLMs can handle symbolic reasoning in purely textual contexts [22, 36, 37], these capabilities remain limited to unimodal inputs, i.e., text. However, many real-world domains, such as autonomous driving, healthcare, law, and finance, demand reasoning that integrates multiple modalities, particularly combining visual and textual information, to support accurate and reliable conclusions. Consider an autonomous driving system that observes a traffic sign (from a camera image) indicating "Road Closed Ahead", given the traffic rule *"Only if the road ahead is open (B), the vehicles may proceed straight (A)."* From the image, the system detects that the road is in fact closed ($\neg B$), and must infer that continuing straight is not permitted ($\neg A$), forming a formal logical reasoning (Modus Tollens; $(A \rightarrow B) \wedge \neg B \rightarrow \neg A$) to avoid traffic accidents. Despite the significance of such multimodal symbolic reasoning, no standard definition or benchmark currently exists for this capability.

To fill this gap, we introduce ***Multimodal Symbolic Logical Reasoning*** (**MuSLR**), a novel task that challenges VLMs to perform symbolic reasoning over combined visual and textual inputs. Figure 1 illustrates the MuSLR task with the above example. We define MuSLR under two task formats: *Truth*

---

[*] Corresponding author: Hao Fei

39th Conference on Neural Information Processing Systems (NeurIPS 2025).

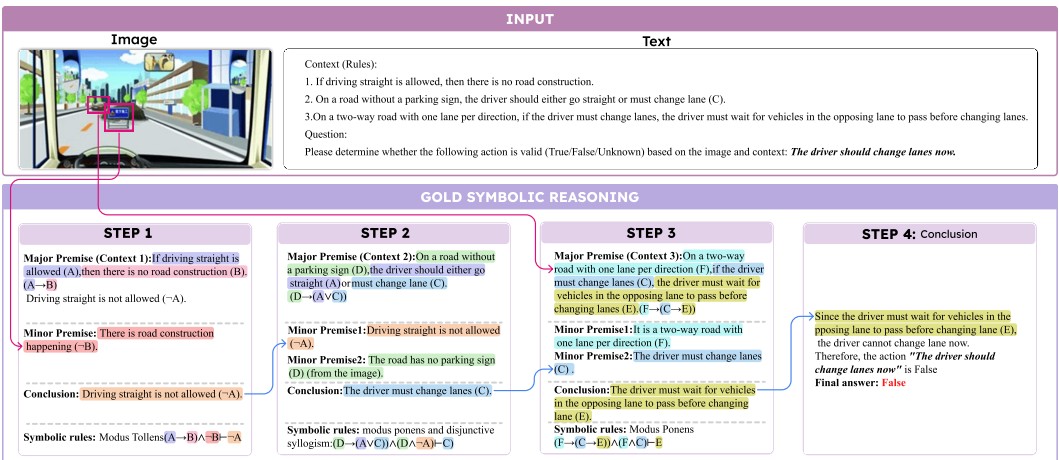

Figure 1: An example of a depth-4 propositional logic task, requiring the VLMs to apply formal symbolic logic rules and integrate multimodalities to reach the conclusion.

*Evaluation* and *Multiple Choice*, where given an image $I$, context $T$, the model must apply symbolic logical reasoning to identify the correct answer. To enable systematic evaluation, we then propose **MuSLR-Bench**, a high-quality benchmark dataset specifically designed to assess the symbolic reasoning abilities of state-of-the-art VLMs. Drawing from authentic web-sourced scenarios where visual and textual content naturally co-occur, we annotate each instance with formal logical rules (e.g., modus ponens) and conduct rigorous quality checks to ensure correctness and logical validity. MuSLR-Bench comprises 1,093 instances spanning 7 domains, including 35 atomic symbolic logic and 976 complex logical compositions, with reasoning depths ranging from 2 to 9 to reflect diverse difficulty levels. In a pilot study, we evaluate seven leading open- and closed-weight VLMs of varying sizes on MuSLR-Bench, revealing that even top models struggle substantially with multimodal symbolic logic inference.

To establish a strong baseline for MuSLR, we further propose **LogiCAM** (**Logi**cal reasoning with **C**ommonsense **A**ugmentation in **M**ultimodalities), which decomposes multimodal symbolic reasoning into modular steps through Chain-of-Thought (CoT) mechanism (cf. Figure 4). First, the `Premise Selector` is designed to address the difficulty of multimodal fusion. We next devise a `Reasoner` module to integrate multimodal evidence and apply symbolic reasoning by approximating formal logical rules, enabling rigorous and systematic deduction to meet the core challenge of MuSLR. Then, the `Reasoning Type Identifier` is designed to address the issue of incomplete information in MuSLR, where heuristics act as supplementary resources to complement symbolic rules when they are insufficient to reach the conclusion. Extensive experiments show that LogiCAM improves GPT-4.1's CoT performance by 14.13% on MuSLR-Bench, achieving even greater gains on complex first-order logic tasks. Further analysis reveals that reasoning performance deteriorates sharply as logical complexity and chain depth increase, highlighting key limitations of current popular VLMs.

In summary, our contributions are fourfold:

- We introduce **MuSLR**, a pioneering task targeting multimodal symbolic logical reasoning, addressing a critical gap in real-world AI reasoning.
- We curate **MuSLR-Bench**, a high-quality dataset comprising 1,093 instances with diverse logical structures and depths, serving as a critical foundation for this topic.
- We develop **LogiCAM**, a strong CoT-based baseline method that decomposes complex reasoning into more manageable and trackable modules.
- Through extensive experiments and analyses, we pinpoint where and why current VLMs struggle with MuSLR, offering insights for future investigation of this area.

## 2   Related Work

**Textual Symbolic Logic Reasoning and Benchmarks.**   Existing benchmarks for symbolic logical reasoning have primarily focused on purely textual settings under formal logic rules. For instance, FOLIO [7] is a human-annotated dataset for complex natural language reasoning equipped with first-order logic annotations to ensure the logical consistency of premises and conclusions. ProofWriter [28] provides small English rulebases of facts and rules with associated questions, requiring models

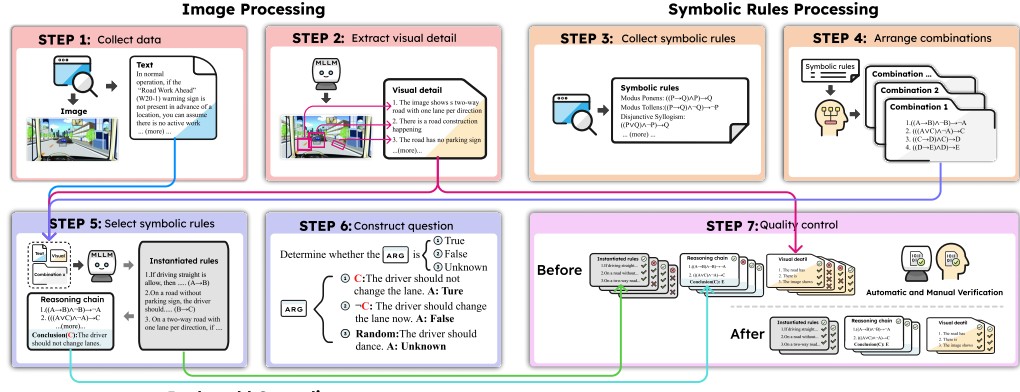

Figure 2: Pipeline of MuSLR data construction. We begin by collecting multimodal data and symbolic rules. These rules are then combined to form reasoning chains, which are grounded in real-world contexts to generate questions and answers, followed by a strict quality check.

to prove or refute statements (or answer "unknown" when proof is impossible) via multi-step natural language proofs. Likewise, Multi-LogiEval [24] evaluates multi-step logical reasoning across propositional, first-order, and even non-monotonic logic types, encompassing over 30 inference rules and various depths to test LLMs' deductive abilities. We further acknowledge numerous additional related works, such as ProntoQA [27], LogicBench [23], and RuleArena [41]. However, these benchmarks assume fully specified, idealized inputs in a single modality (text) and do not incorporate visual information, limiting their direct applicability to real-world scenarios.

**Multimodal Reasoning and Benchmarks.** In parallel, several benchmarks have introduced accessing reasoning in vision and language [5, 33]. LogicVista [35] evaluates VLMs' logical reasoning in visual contexts, with 448 annotated multiple-choice questions spanning a spectrum of logical reasoning tasks and capabilities. Similarly, VisuLogic [38] targets vision-centric reasoning by constructing tasks that require robust visual logic without relying on textual descriptions or shortcuts. Meanwhile, broader vision-language benchmarks emphasize contextual reasoning rather than formal logic: for example, MMMU [39] offers college-level multimodal questions across six disciplines (e.g., charts, maps, chemical structures), testing domain-expert reasoning. MathVista [18] targets compositional mathematical inference in visual scenarios. However, none of these multimodal benchmarks explicitly test the application of formal logical rules (e.g. Modus Ponens or De Morgan's Law) grounded in both visual and textual input. **MuSLR** addresses this gap by requiring explicit symbolic logical deduction from joint visual–textual inputs, integrating formal logic rules into multimodal understanding.

**Neuro-Symbolic Reasoning Method.** Many prior works adopt a symbolic prover in the reasoning pipeline to achieve rigorous and reliable reasoning. Typically, an LLM is used to formalize natural language into symbolic form, after which a theorem prover is employed to solve it [11, 19, 22, 26, 34]. However, theorem provers only accept text input. In multimodal scenarios, this requires first converting visual or multimodal information into text, a process that inevitably leads to information loss and thus limits adaptability. In contrast, our LogiCAM framework is designed to approximate symbolic reasoning using a vision–language model (VLM), which has direct access to multimodal information without relying on lossy translation.

## 3 Task Definition

The proposed tasks require models to integrate information from both an image $I$ and a text passage $T$ to perform reasoning, ensuring that neither modality alone is sufficient for correct inference. The tasks explicitly emphasize **multimodal reasoning**, where the fusion of visual and textual context is essential for deriving accurate and consistent conclusions.

**Task-I: Truth Evaluation (True/False/Unknown) Question.** Given an image $I$, a text passage $T$, and an argument $A$, the model must determine the truth value of the argument based on the combined information from $I$ and $T$. Specifically, the model outputs the truth value $\text{Truth}(A) \in \{\text{True}, \text{False}, \text{Unknown}\}$ and generates a sequence of reasoning steps $R = \{R_1, R_2, \ldots, R_n\}$, where each $R_i$ represents an individual step that contributes to the final decision. Formally, the input is a triplet $(I, T, A)$, and the output consists of $\text{Truth}(A)$ and $R$.

| Statistics | Numbers |
|---|---|
| Total instances | 1093 |
| Total sources | 7 |
| Domain (#) | 7 |
| Symbolic logic (#) | 3 |
| Atomic symbolic rules (#) | 35 |
| Symbolic rule combination (#) | 976 |
| Min reasoning depth | 2 |
| Max reasoning depth | 9 |
| Min context length | 35 |
| Max context length | 1484 |
| Avg. context length | 554.9 |

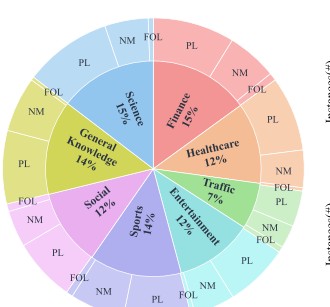
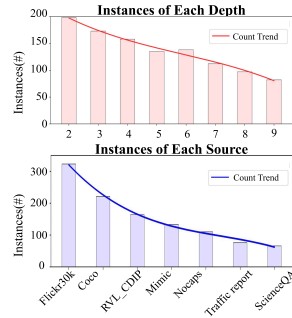

Figure 3: Dataset Statistics. The left table presents general dataset statistics. The middle pie chart illustrates the distribution across domains and symbolic logic. The right bar charts display the number of instances by reasoning depth and data source.

**Task-II: Multiple Choice Question.** Given an image $I$, a text passage $T$, and candidate arguments $\{A_1, A_2, A_3, A_4\}$, the model must select the argument that best matches the image and text, denoted as $\text{BestArgument}(I, T) \in \{A_1, A_2, A_3, A_4\}$. Additionally, the model must provide detailed reasoning steps $R = \{R_1, R_2, \ldots, R_n\}$, where each $R_i$ details a step in the reasoning process. Formally, the input is a triplet $(I, T, \{A_1, A_2, A_3, A_4\})$, and the output consists of $\text{BestArgument}(I, T)$ and $R$.

## 4  MuSLR-Bench: A Benchmark for Multimodal Symbolic Logical Reasoning

**Dataset Construction.** We collect images from various sources such as COCO [14], Flickr30k [25], nocaps [1], Mimic [10], RVL_CDIP [8], ScienceQA [17], and manually collected Traffic Reports. Visual details for each image are extracted using GPT-4o, ensuring diverse and fine-grained descriptions. We carefully select non-trivial logical inference rules, such as Modus Ponens and Hypothetical Syllogism, drawn from propositional logic (PL), first-order logic (FOL), and non-monotonic logic (NM). These rules then form meaningful but abstract reasoning chains through logical combinations. The abstract chains are grounded in real-world contexts by leveraging extracted visual features and relevant retrieved text from sources like healthcare, traffic reports, and Wikipedia. Questions and answers are then generated based on these instantiated reasoning chains, using rule-based substitution.

To ensure the quality and relevance of the dataset, both automatic and manual quality control procedures are employed. Automatic checks include assessing lexical similarity and commonsense plausibility, while human annotators verify the accuracy of visual details and the real-world relevance of the generated context. Instances that fail these checks are filtered out, ensuring a high-quality, logically sound, and contextually relevant dataset. Further details on the data construction and quality control processes are provided in the Appendix B and C, respectively.

### 4.1  Dataset Highlights

MuSLR consists of 1093 instances, where each instance includes a multimodal context (image and associated text), a ground-truth logical reasoning chain, and corresponding question-answer pairs. The dataset is constructed to support both detailed symbolic logical reasoning analysis and challenging multimodal reasoning tasks. Below, we summarize the key features of the dataset:

**Ground-Truth Reasoning Steps.** Each instance is equipped with an explicit, step-by-step ground-truth reasoning chain, enabling detailed analysis and training of models for symbolic logical reasoning.

**Multi-Scenario Coverage.** The dataset spans a wide range of domains, including science, entertainment, sports, social issues, general knowledge, traffic, healthcare, and finance. The distribution across these scenarios is illustrated in the pie chart in Figure 3.

**Diverse Symbolic Reasoning Types.** MuSLR contains diverse symbolic logic: propositional logic (PL), first-order logic (FOL), and non-monotonic logic (NM), ensuring broad logical coverage.

**Multimodality.** To the best of our knowledge, this is the first dataset that combines both image and text modalities for symbolic logical reasoning tasks grounded in formal logical rules.

**Diverse Difficulty Levels.** The reasoning chains vary in depth from 2 to 9 steps, offering a broad spectrum of difficulty levels and supporting evaluation across simple and complex reasoning scenarios.

**Multiple Question Types.** The dataset supports multiple question formats, including **Truth Evaluation** and **Multiple-Choice** questions, allowing for diverse model evaluation protocols.

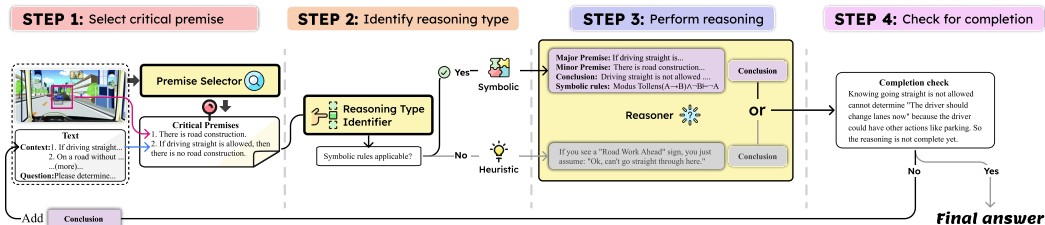

Figure 4: **LogiCAM** Workflow. The figure illustrates a single iteration; the complete multi-iteration reasoning process is detailed in Section 9.

## 4.2 Challenge

MuSLR presents five key challenges for developing robust multimodal symbolic reasoning models:

**Integrate Multimodality.** Can the model extract and integrate critical visual and textual context to construct valid reasoning chains? (See Section 7.5)

**Step-by-Step Symbolic Reasoning Tracability.** Can the model produce interpretable, verifiable, step-by-step reasoning processes in valid logic? (See Section 7.2)

**Blend Heuristics for Symbolic Reasoning.** Can the model apply heuristic reasoning when symbolic logic is insufficient?

**Diverse Symbolic Logic.** Can the model handle various forms of symbolic logic (PL, FOL, and NM)? (See Section 7.1)

**Reasoning Depth Handling.** Can the model reason over different depths, maintaining consistency in longer chains? (See Section 7.3)

Addressing these challenges requires models to integrate multimodal perception and systematic logical reasoning, thereby providing a solid foundation for advancing multimodal reasoning systems.

## 5 LogiCAM: A Modular MuSLR Framework

We propose a modular framework, **LogiCAM** (**Logi**cal reasoning with **C**ommonsense **A**ugmentation with **M**ultimodality), which consists of three modules based on GPT-4.1, as illustrated in Figure 4. Each module is designed to address a specific challenge posed by MuSLR. The modules work together to solve different problem components, which include: (1) the *Premise Selector*, (2) the *Reasoning Type Identifier*, and (3) the *Reasoner* module. Below, we explain how each module addresses its challenge and contributes to the reasoning chain.

**Select Critical Multimodal Premises.** The *Premise Selector* is designed to address the multimodalities integration challenge, which involves the need to process both visual and textual data to extract critical premises. Given an image $I$ and textual information $T$ containing context $\mathcal{T}$ and question $Q$, this module directs the VLM to first select the most relevant symbolic rules $R_r \in \mathcal{T}$. The VLM will then analyze the symbolic logic $R_r$ to determine which part is relevant to the image and extract the corresponding visual information $V_r$. In this way, the system ensures that only the most critical visual and textual details are extracted, avoiding unnecessary complexity and noise from abundant data. The symbolic rule $R_r$ and visual details $V_r$ will be combined and denoted as $I_{\text{critical}}$.

**Identify Reasoning Type.** The *Reasoning Type Identifier* addresses the blend of heuristics and symbolic, which involves determining whether symbolic reasoning or heuristics should be applied during each reasoning iteration. The core challenge is deciding when symbolic logic is sufficient and when heuristics should be used to complement symbolic reasoning. To solve this, the *Reasoning Type Identifier* analyzes the selected premises $I_{\text{critical}}$ and determines whether formal logical rules can be applied. If so, prioritize it. Otherwise, heuristics and commonsense reasoning are employed to compensate for the limitations of purely symbolic reasoning. In this way, the model maximizes the rigor and soundness of the reasoning by prioritizing symbolic reasoning while maintaining flexibility to supplement additional knowledge through commonsense-driven heuristics when symbolic reasoning alone is insufficient.

**Perform Reasoning.** The *Reasoner* is central to addressing symbolic reasoning tracability, which uses a VLM to approximate formal logical rules when symbolic reasoning is required. Depending on the outcome of the Reasoning Type Identifier, the *Reasoner* either applies symbolic reasoning or uses heuristic commonsense to complete the reasoning process. If *symbolic reasoning* is selected,

| Model | Symbol | Healthcare | Traffic | Sports | Ent. | Social | Science | Finance | General |
|---|---|---|---|---|---|---|---|---|---|
| *Three-shots CoT Open-Weight VLMs* | | | | | | | | | |
| Qwen | PL | 50.00 | 33.33 | 33.33 | 42.67 | 36.49 | 48.54 | 54.17 | 46.51 |
| | FOL | 0.00 | 42.86 | 50.00 | 40.00 | 66.67 | 16.67 | 22.22 | 25.00 |
| | NM | 43.18 | 25.93 | 43.75 | 35.42 | 23.26 | 43.75 | 54.24 | 37.50 |
| Llava | PL | 20.45 | 30.95 | 37.18 | 32.00 | 22.97 | 34.95 | 27.08 | 43.02 |
| | FOL | 0.00 | 57.14 | 50.00 | 20.00 | 44.44 | 66.67 | 55.56 | 50.00 |
| | NM | 31.82 | 37.04 | 45.31 | 43.75 | 39.53 | 47.06 | 25.42 | 45.31 |
| InternVL | PL | 57.95 | 42.86 | 37.97 | 44.00 | 37.84 | 46.60 | 51.04 | 50.00 |
| | FOL | 50.00 | 42.86 | 50.00 | 20.00 | 66.67 | 50.00 | 22.22 | 50.00 |
| | NM | 38.64 | 29.63 | 46.88 | 35.42 | 46.51 | 49.02 | 45.76 | 43.08 |
| InstructBlip | PL | 42.05 | 33,33 | 39.2 | 26.67 | 36.49 | 29.13 | 40.62 | 25.58 |
| | FOL | 50.00 | 28.57 | 25.00 | 40.00 | 55.56 | 16.67 | 22.22 | 25.00 |
| | NM | 52.27 | 40.74 | 53.12 | 31.25 | 44.19 | 35.29 | 2.34 | 30.77 |
| *Three-shots CoT Closed-Weight VLMs* | | | | | | | | | |
| Claude | PL | 44.32 | 26.19 | 24.36 | 26.67 | 28.38 | 35.92 | 36.46 | 34.88 |
| | FOL | 50.00 | 14.29 | 50.00 | 20.00 | 55.56 | 0.00 | 44.44 | 75.00 |
| | NM | 29.55 | 37.04 | 32.81 | 29.17 | 30.23 | 43.14 | 38.98 | 31.25 |
| GPT-4o | PL | 45.45 | 40.48 | 33.33 | 37.50 | 34.72 | 37.00 | 28.99 | 43.90 |
| | FOL | 0.00 | 14.29 | 25.00 | 0.00 | 37.50 | 33.33 | 50.00 | 33.33 |
| | NM | 52.27 | 48.15 | 35.48 | 50.00 | 52.38 | 45.10 | 41.46 | 32.81 |
| GPT-4.1 | PL | 54.55 | 50.00 | 44.30 | 41.33 | 33.78 | 43.69 | 45.83 | 51.16 |
| | FOL | 0.00 | 14.29 | 50.00 | 20.00 | 44.44 | 16.67 | 33.33 | 0.00 |
| | NM | 47.73 | 59.26 | 46.88 | 50.00 | 53.49 | 56.86 | 61.02 | 40.62 |
| **LogiCAM** | PL | 63.64 (+9.09) | 61.90 (+11.90) | 58.23 (+13.93) | 64.00 (+22.67) | 56.76 (+22.98) | 57.28 (+13.59) | 53.68 (+7.85) | 67.44 (+16.28) |
| | FOL | 50.00 (+50.00) | 60.42 (+46.13) | 50.00 (+0.00) | 60.00 (+40.00) | 44.44 (+0.00) | 40.00 (+23.33) | 75.00 (+41.67) | 75.00 (+75.00) |
| | NM | 63.64 (+15.91) | 66.67 (+7.41) | 58.23 (+11.35) | 60.42 (+10.42) | 74.42 (+20.93) | 64.71 (+7.85) | 74.14 (+13.12) | 55.38 (+14.76) |

Table 1: Main Results. Blue indicates the best open-weight VLM, and Red indicates the best closed-weight VLM. The (red brackets) indicate our improvement over the base model.

the module applies formal logical rules to the premises $I_{\text{critical}}$ and derives a conclusion $C$ based on a syllogism, which draws a result from the major and minor premises. This reasoning process ensures that conclusions are drawn according to sound logical principles. If *heuristics* are selected, the module uses commonsense reasoning to bridge gaps left by symbolic logic. This design makes sure that the model can perform symbolic reasoning grounded in logical principle, while relax this restriction when heuristics are required. A full example can be found in the Figure 9.

**Check for Completion.** Finally, the system checks whether the conclusion $C$ is sufficient to answer the question $Q$. If so, it concludes the final answer. Otherwise, the system appends the conclusion $C$ to the context $\mathcal{T}$, resulting in $T' = T \cup C$, and starts over the whole reasoning iteration.

# 6 Experiments

## 6.1 Settings

**Evaluation.** We evaluate models based on two dimensions: direct answer match and reasoning accuracy. Direct answer match measures the correctness of the final answer, while reasoning accuracy evaluates the quality of the step-by-step reasoning. Reasoning accuracy is computed by comparing model-generated steps with ground-truth steps using ROUGE-L [13] and BertScore-F1 [40]. We also assess ROSCOE [6], which measures logical coherence, factual grounding, and informativeness step by step. More details are in Section 7.2.

**Baseline.** For benchmarking, we consider multiple state-of-the-art models. For open-source models, we benchmark `Qwen2.5-VL-7B-Instruct` [3], `Llava-1.5-7B` [15], `InternVL3-8B` [42], and `Instructblip-Vicuna-13B` [4]. For closed-source models, we evaluate `GPT-4o` [20], `GPT-4.1` [21] and `Claude-3.7-Sonnet` [2]. These models are chosen to represent the current SoTA in multimodal reasoning.

**Settings.** To ensure reproducibility, all models are evaluated under standardized settings. We adopt a three-shot Chain-of-Thought (CoT) [32] prompting setup. For language model sampling, the temperature is set to 0.0 to minimize randomness and encourage deterministic outputs.

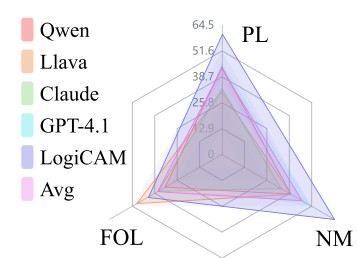

Figure 5: Accuracy of symbolic logic

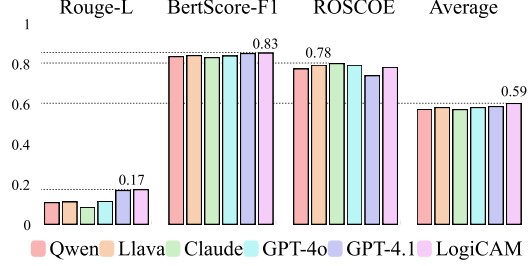

Figure 6: Comparison of models' reasoning tracability

## 6.2 Main Results and Observations

The main results are presented in Table 1. We have the following observations:

**Closed-weight models generally outperform, but open-weight models can rival or surpass them.** GPT-4.1 leads with 46.84%, followed closely by InternVL at 45.20%, the top open-weight model. Qwen (41.63%) and GPT-4o (38.93%) follow in the second tier, with InstructBLIP (35.59%), Llava (35.13%), and Claude (33.49%) at the lower end. The performance gap between top and bottom is just 13.35%. These results show that while closed-weight models typically excel, well-designed open-weight models can sometimes outperform proprietary models

**LogiCAM enhances CoT and achieves the highest overall performance, with especially strong gains in complex symbolic logic.** Integrating LogiCAM into GPT-4.1 results in a substantial performance boost, increasing the average accuracy by 14.13%. When examined by logic type, the improvements are consistent yet differ in scale: FOL accuracy increases by 48.93 %, PL by 31.93 %, and NM by 26.17 %. This pattern indicates that the advantage of LogiCAM grows with the complexity of the logic type: the largest relative improvement is observed in FOL, the most structurally demanding form, followed by PL, and then NM, which is more aligned with intuitive human reasoning and less dependent on rigid symbolic structure. These results suggest that LogiCAM not only strengthens general symbolic reasoning but is especially effective in complex logical operations.

## 7 Analysis and Discussion

We conduct additional experiments and perform detailed analysis to gain deeper insights into the multimodal symbolic reasoning capabilities of current VLMs.

### 7.1 Effects on Different Types of Symbolic Logic

In Figure 5, we evaluate the accuracy of each symbolic logic and found that **Model accuracy decreases with rising symbolic complexity: VLMs perform best with non-monotonic reasoning, less well with propositional logic, and struggle most with first-order logic.** First-order logic has the lowest average accuracy at 37.04%, due to its strictest formalism and need for precise variable binding and quantifier tracking. Propositional logic fares better with 42.77%, as its simpler structure eases syntactic constraints. Non-monotonic reasoning performs best at 46.09%, due to its closer alignment with human cognition and requiring less rigid symbolic manipulation. Overall, as symbolic complexity increases, model accuracy declines, highlighting the challenges of fine-grained logical abstraction in current VLMs.

### 7.2 Tracability of Reasoning Step

As shown in Figure 6, LogiCAM leads in both ROUGE-L (0.170) and BertScore (0.835), with the highest overall mean (0.590), indicating its outputs closely match human phrasing and meaning. Claude scores highest on ROSCOE (0.784), reflecting strong logical consistency but performs poorly on ROUGE-L (0.084). GPT-4.1 balances phrasing and semantics (ROUGE-L = 0.166%, BertScore = 0.833%) but shows moderate stepwise justification (ROSCOE = 0.725%), suggesting occasional logical gaps. Llava and GPT-4o have similar profiles (Average = 0.570%), demonstrating that strong semantic similarity (0.822%) doesn't guarantee superior inference quality (ROSCOE = 0.776%).

**Surface-level or semantic objectives alone don't ensure logical coherence. Future work should include logic-focused training goals.** A Pearson's correlation analysis reveals a weak correlation between ROUGE-L and ROSCOE ($r = 0.25$) but a moderate correlation between BertScore and ROSCOE ($r = 0.65$), suggesting that surface-level metrics do little for logical coherence, while semantically rich training helps more. Claude's high ROSCOE but low ROUGE and BertScore

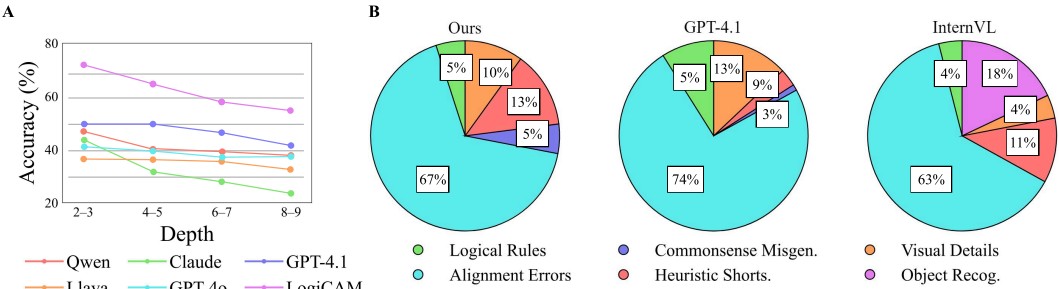

Figure 7: Panel A reports accuracy across different depths, while Panel B illustrates the error distribution across models.

highlights that reasoning-focused objectives improve logical rigor, often at the cost of natural phrasing. This suggests that optimizing for surface or semantic metrics alone isn't enough to improve logical coherence, and future research should target the quality of symbolic logic.

### 7.3 Depth Analysis

As shown in Figure 7A, **All models exhibit a clear decline in performance as the symbolic reasoning depth increases**, confirming the benchmark's effectiveness in exposing the growing complexity of multimodal logical tasks. GPT-4.1 emerges as the strongest baseline, with the highest accuracy after LogiCAM and a moderate 16% drop from 2–3 to 8–9 steps. However, it still struggles at greater depths, revealing limits in complex multi-hop reasoning. GPT-4o and Llava maintain stable performance with minor 3–4% drops, but their overall accuracy is much lower, indicating a trade-off between robustness and reasoning capacity. In contrast, Claude suffers a sharp 20% decline, highlighting poor generalization on longer symbolic chains.

**In contrast, LogiCAM not only delivers superior average performance but also scales more effectively when reasoning chains grow.** It demonstrates the strongest overall performance and robustness, consistently outperforming other models across all reasoning depths. It achieves 71.91% accuracy at the shallowest level and maintains a solid 54.61% even at the deepest. Notably, it surpasses the strongest baseline GPT-4.1 by 13% at depths 8–9, highlighting a substantial advantage in handling extended reasoning chains. While LogiCAM exhibits a larger absolute drop across depths, its high performance at all levels indicates strong generalization to both moderate and complex symbolic reasoning tasks. This drop, however, suggests there is still room to improve long-chain reasoning robustness.

### 7.4 Ablation Study

We conduct an ablation study, which demonstrates that each module is indispensable, as shown in Figure 8A. Removing the symbolic reasoning module produces the largest performance reduction (5.14%), underscoring the importance of adhering to formal logical rules. Omitting heuristic reasoning yields a 3.45% degradation, indicating that heuristics serve as an effective complement when strict logical rules are inapplicable. Disabling premise selection results in a 3.27% drop, reflecting its crucial role in identifying critical information and simplifying subsequent inference. Collectively, these findings highlight that each module plays a critical and non-redundant role, underscoring the necessity of the full design for achieving strong overall performance.

### 7.5 Error Analysis

We conduct a thorough error analysis by randomly selecting a domain- and symbol-balanced subset of 100 examples for each model. We identify six major error types: incorrect application of logical rules, failure to supplement with heuristic commonsense knowledge, overlooking critical visual details, logical misalignment between visual and textual context, improper reliance on heuristic shortcuts where symbolic reasoning is required, and misperception of objects in the image. Details of each error type are discussed in Appendix A.

**Error distribution across different models.** As shown in Figure 7 B, failures to logically align and integrate visual with textual premises overwhelmingly dominate (67% for LogiCAM, 74% for GPT-4.1, and 63% for InternVL), demonstrating that cross-modal grounding remains the principal hurdle. Looking specifically at each model:

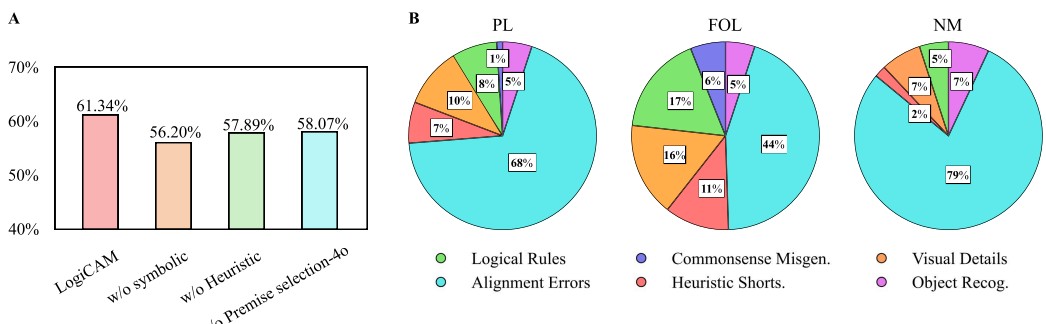

Figure 8: Panel A shows the ablation study results as bar plots, while Panel B presents pie charts illustrating the error distribution across different logical types.

- **LogiCAM** is designed to blend symbolic deduction with heuristic inference; it exhibits a high rate of heuristic shortcuts (13%), indicating difficulty in discerning when to apply formal logic versus commonsense reasoning.
- **GPT-4.1** shows minimal reliance on heuristics (3%) and almost no failures to supplement with commonsense (1%), yet overlooks visual details in 13% of cases and misapplies formal logical rules 9% of the time. The latter aligns with known Chain-of-Thought behavior, where outputs can seem plausible but contain subtle logical errors [37].
- **InternVL** suffers the highest proportion of pure perception errors (18%), reflecting weaker object recognition than GPT-4.1, and relies on heuristic shortcuts in 11% of cases.

Notably, all models suffer major logical misalignment between modalities and visual oversight errors, underscoring a critical need for advances in vision–language fusion. Future work should focus on improving cross-modal fusion and incorporating logic-based training objectives, enabling more accurate symbolic reasoning across modalities.

**Error distribution across different logical types.** We further analyze the error by logical types as shown in Figure 8B, and have the following findings:

- **Consistent Alignment Issues Across Logic Types.** A primary source of failure in PL, FOL, and NM arises from logical misalignment between text and image, with this problem being particularly severe in NM (79%) and PL (68%). This aligns with our broader finding that mapping formal logical structures onto multimodal contexts remains a fundamental challenge for current vision-language models (VLMs).
- **FOL is Most Prone to Overlooking and Logical Errors.** Overlooking errors are most frequent in FOL (16%), where models often miss details in multi-entity, nested, or quantified reasoning. Logical rule errors are also highest (17%), reflecting the symbolic complexity of quantifier binding, variable tracking, and relational reasoning compared to PL or NM.
- **PL's Dependence on Symbolic Alignment.** Although PL avoids many deep logical errors, its performance is highly dependent on accurate logical text-image alignment, as reflected in the 68% rate of alignment errors. Once alignment is achieved, the relatively simple structure of PL facilitates more reliable rule application by the models.
- **NM's High Alignment Difficulty but Low Logical Error Rates.** Despite exhibiting the highest rate of alignment errors (79%), NM shows the lowest incidence of incorrect logical rule application (5%) and commonsense supplementation errors (0%). This pattern suggests that once alignment is successfully established, NM reasoning is more consistent with the model's intuitive understanding or default interpretive patterns, which may partly explain its comparatively strong raw performance.

### 7.6 Case Study

To illustrate the limitations of existing VLMs and how LogiCAM addresses them, we present a case study comparing the reasoning of GPT-4.1 (with CoT prompting) and LogiCAM in Figure 9.

**GPT-4.1's CoT reasoning exhibits a form of "nearsightedness".** As the reasoning chain grows longer, it gradually loses the thread that connects image cues to abstract premises, defaulting instead to surface-level judgments (e.g., "I can't see a predator, so unknown"). Without a systematic **Premise Selection** process, it fails to ground observations like "on grass" in relevant textual logical rules (e.g., (not on the grass (A) or searching for food (B)) ∧ (on the grass (¬A)) →

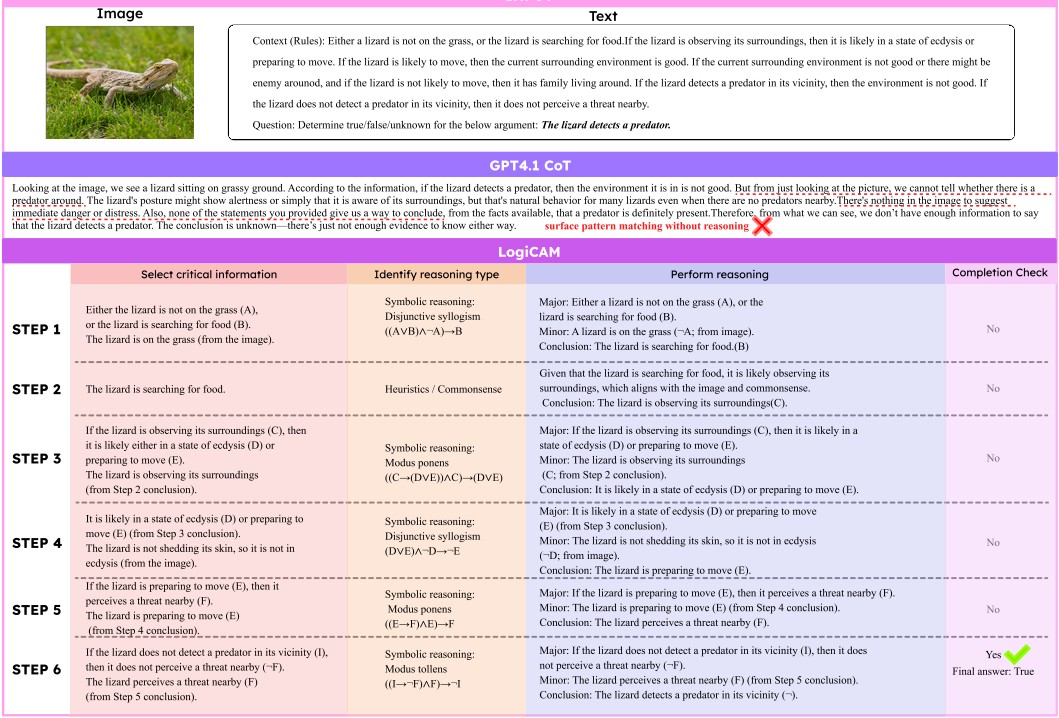

Figure 9: A Case Study Comparing CoT and LogiCAM

searching for food (B). Moreover, lacking step-by-step formal inference, it eventually abandons the deeper reasoning chain altogether, falling back to superficial pattern matching.

**In contrast, LogiCAM systematically derives new knowledge and reaches the correct answer by integrating three tightly-coupled mechanisms at every inference step.** Its **Premise Selection** module continuously extracts and logically maps image features into textual element (e.g., "on grass" → food search; "no shedding" → ¬ecdysis), demonstrating its advantages in multimodal fusion. The **Reasoning Type Identifier** then selects the appropriate reasoning type, formal logic for structured inferences (e.g., $C \rightarrow (D \vee E)$) or heuristics to complement symbolic logic, thereby balancing the rigor of formal deduction with the flexibility to incorporate knowledge beyond the scope of logic. Finally, the **Symbolic Reasoner** rigorously applies formal inference rules (e.g., disjunctive syllogism, modus ponens, modus tollens) to derive each new conclusion in a systematic and reliable way. This disciplined, iterative process ensures robustness in handling long reasoning chains.

# 8 Conclusion and Future Work

We have pioneered the **Multimodal Symbolic Logical Reasoning (MuSLR)** task, challenging models to perform precise, rigorous formal logic inferences over combined visual and textual inputs, thereby filling a critical gap in existing benchmarks. To support this research direction, we release **MuSLR-Bench**, a rigorously annotated dataset of 1,093 instances spanning seven application domains, featuring 35 atomic reasoning units and 976 composite logic combinations with depths ranging from 2 to 9. We also propose a strong baseline **LogiCAM**, a novel modular framework that systematically decomposes the reasoning process into premise selection, reasoning-type identification, and formal inference, demonstrating substantial performance gains over prior methods.

Looking forward, our diagnostic analyses reveal two key opportunities for advancing multimodal symbolic reasoning. First, **integrating dedicated symbolic modules is essential**: the LogiCAM outperforms base VLMs precisely because it extracts multimodalities based on logic and embeds explicit symbolic reasoning steps. Second, **existing VLMs struggle to align and fuse visual and textual information when performing formal logic**; Future work should explore tighter multimodal integration, such as cross-modal architectures trained with logic-grounded objectives, to bridge this gap. By making MuSLR and its benchmark publicly available, we hope to catalyze research on these challenges and bring truly rigorous, multimodal symbolic reasoning within reach.

## Acknowledgments and Disclosure of Funding

This work is supported by the Ministry of Education, Singapore, under its MOE AcRF TIER 3 Grant (MOE-MOET32022-0001).

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

# Appendix

In the appendix, we provide detailed descriptions of each error type (Section A), the complete workflow of the MuSLR construction pipeline (Section B), the full quality control process, including both automatic and manual filtering strategies (Section C), the details of the LogiCAM framework (Section D), the collection of atomic symbolic logic used in our study (Section F), and an ethics statement (Section G).

## A  Error Analysis

We provide detailed explanations of each error type below.

**Incorrect Application of Logical Rules**   This error occurs when the model attempts to apply formal logical rules but does so incorrectly. Typical mistakes include reversing implications, confusing necessary and sufficient conditions, or failing to properly follow multi-step deductions. While the model recognizes that logical reasoning is needed, the specific application is flawed, leading to invalid conclusions.

**Failure to Supplement with Commonsense / Rule Misgeneralization**   In some cases, the given input lacks complete information, requiring the model to draw on commonsense knowledge to fill in gaps. This error happens when the model fails to do so, resulting in halted or incomplete reasoning. Alternatively, the model may overgeneralize a formal rule, applying it too broadly or narrowly, which also leads to incorrect outcomes.

**Overlooking Visual Details**   This error reflects the model's inability to notice or correctly interpret critical visual elements in the image, such as small objects, specific colors, or spatial relationships. Missing these details prevents the model from correctly progressing in its reasoning chain, despite the necessary information being present in the visual input.

**Premise Integration / Alignment Errors**   Even when the model successfully extracts information from both text and image, it sometimes fails to align them correctly. This happens when visual entities are mismatched with their textual references (e.g., linking "the red triangle" to the wrong object in the image). Such misalignment breaks the reasoning process and leads to incorrect answers.

**Heuristic Shortcuts over Formal Logic**   Rather than following precise logical reasoning, the model occasionally defaults to heuristic-based shortcuts, relying on superficial patterns or associations learned during training. While this may sometimes produce plausible answers, it undermines the rigor required for formal logical tasks, resulting in systematic errors when heuristics are misapplied.

**Visual Perception / Object Recognition Errors**   This error type stems from failures in basic visual perception, such as misidentifying objects, misclassifying shapes, colors, or spatial positions. When the model starts reasoning from an incorrect visual premise, all subsequent deductions are built on a faulty foundation, leading to incorrect conclusions.

## B  MuSLR Construction Process

We collect images from multiple sources, including COCO [14], Flickr30k [25], nocaps [1], Mimic [10], RVL_CDIP [8], ScienceQA [17] and Traffic Report collected manually. For each image $I$, visual details $V$ are extracted using GPT-4o to ensure diverse and fine-grained descriptions.

**Step 1: Systematic Rule Selection**

We begin by examining a broad set of logical inference rules drawn from propositional logic (PL), first-order logic (FOL), and non-monotonic logic (NM). We utilize the complete set of logical rules collected by [24], denoted as $\mathcal{R} = \{r_1, r_2, \ldots, r_m\}$, which comprehensively covers standard inference patterns. Rather than selecting rules randomly, we carefully curate a subset $\mathcal{R}_{\text{selected}} \subseteq \mathcal{R}$ that is both formally sound and frequently encountered in real-world reasoning. This subset includes classical patterns such as Modus Ponens, Hypothetical Syllogism, Modus Tollens, and Disjunctive Syllogism. Details about the logical rules are provided in the Appendix.

**Step 2: Meaningful Rule Composition:**

We select meaningful rule combinations, denoted as $\mathcal{R}_{\text{set}} = \{R_1, R_2, \ldots\}$, to construct logically coherent reasoning chains $\mathcal{C} = \{C_1, C_2, \ldots\}$ by rule-based substitution. Each reasoning chain $C_i$

consists of an ordered sequence of rules from $\mathcal{R}_{\text{set}}$ and is manually composed by experts in formal logical reasoning to ensure coherence and meaningfulness.

**Step 3: Grounding in Real-World Contexts:**

The meaningful rule composition step produces an abstract, context-independent symbolic rule set $\mathcal{R}_{\text{set}} = \{R_1, R_2, \ldots\}$ (e.g., "If $A$, then $B$"). During grounding, visual features $V$ from an image $I$ guide the retrieval of relevant textual information $T_{\text{retrieved}}$ from sources like healthcare reports, Wikipedia, or traffic incident summaries. Abstract rules from $\mathcal{R}_{\text{set}}$ are instantiated using real-world information from $T_{\text{retrieved}}$, creating the grounded rule set $\mathcal{R}_{\text{real}}$ (e.g., "If someone is blowing out candles, they might be celebrating a birthday").

The adapted rule set $\mathcal{R}_{\text{real}}$ will be used to construct the instantiated reasoning chain $\mathcal{C}_{\text{real}}$. When the symbolic reasoning rule $\mathcal{R}_{\text{real}}$ alone is insufficient to capture the real-world context $T_{\text{retrieved}}$, we incorporate commonsense reasoning to supplement formal logic. This combination forms a hybrid reasoning structure $\mathcal{C}_{\text{hybrid}} = (r_1, r_2, \ldots, r_k)$, where each $r_i \in \mathcal{R}_{\text{sym}} \cup \mathcal{R}_{\text{cs}}$. Here, $\mathcal{R}_{\text{sym}}$ comprises rules instantiated from $\mathcal{R}_{\text{set}}$, and $\mathcal{R}_{\text{cs}}$ denotes commonsense reasoning steps. Commonsense reasoning is incorporated only in $\mathcal{C}_{\text{hybrid}}$ and not explicitly represented in $\mathcal{R}_{\text{real}}$. This reflects human cognitive processes, where not all necessary information is always available, and intuitive reasoning is often used to fill in the gaps. The $\mathcal{R}_{\text{real}}$ populates the hybrid reasoning template $\mathcal{C}_{\text{hybrid}}$, yielding the fully grounded reasoning chain $\mathcal{C}_{\text{real}}$. Then we use the conclusion of the $\mathcal{C}_{\text{real}}$ to construct questions and ground-truth answers based on rule-based substitution.

**Step 4: Question Generation**

Based on the ground-truth reasoning chain $C_{\text{gt}}$ and answer $A_{\text{gt}}$, we generate corresponding questions $Q$ that require multi-step reasoning for solution, following rule-based substitution templates.

**Step 5: Automatic and Manual Quality Verification**

Finally, both automatic verification procedures and manual expert review are employed to ensure the overall quality, consistency, and correctness of the generated dataset.

# C MuSLR Quality Check

To ensure the high quality, relevance, and correctness of the constructed dataset, we implement a multi-layered quality control procedure combining both automatic and manual verification steps.

**Automatic Quality Control:** We apply two automatic filtering strategies to enforce logical soundness and diversity:

- **Lexical Similarity Filtering:** We compute the lexical similarity between each pair of reasoning steps within a reasoning chain using Jaccard Similarity. Chains with a similarity score above $0.5$ are discarded to promote step diversity and minimize redundancy.

- **Commonsense Plausibility Filtering:** Each reasoning step is assessed using Vera [16], a T5 model fine-tuned on commonsense reasoning tasks. If any step receives a plausibility score below $0.5$, the entire instance is removed to ensure logical soundness and realism.

As a result of the automatic filtering, the original sample size was reduced from 1,956 to 1,464.

**Manual Quality Control:** Given that the extraction of visual details ($V$) leverages GPT-4o, which may have hallucinations, we implement a rigorous manual validation stage:

- **Visual Detail Verification:** Human annotators confirm that the extracted visual details accurately reflect the content of the corresponding image, explicitly checking for hallucinated objects, actions, or attributes.

- **Context and Question Evaluation:** Annotators evaluate whether the generated context ($T_{\text{context}}$) and associated questions ($Q$) are plausible and relevant to real-world scenarios.

**Annotation Process and Training** All instances were independently reviewed by three trained annotators with STEM backgrounds. In total, six annotators were recruited to assess the 1,464 instances, with each annotator reviewing 732 instances. For each check, annotators provided judgments using a three-option scale: Yes, No, or Not Sure.

To prepare annotators and ensure consistent application of quality standards, we provided a dedicated training session. This session covered task definitions, annotation guidelines, and hands-on practice with feedback. To further support annotators and minimize cognitive load, we developed a custom annotation interface prototype (see Figure 10), which streamlined the annotation process by integrating

image previews, visual details, and context input fields for both checks. This tool helped reduce annotation errors and improve task efficiency.

Annotators also underwent a calibration phase involving 30 examples, followed by iterative discussion sessions to refine annotation guidelines and resolve disagreements. We measured inter-annotator agreement using **Fleiss' Kappa**, achieving an average score of 0.92 for visual detail verification (substantial agreement) and 0.71 for context alignment (moderate agreement), which is consistent with the subjective complexity of evaluating real-world plausibility.

**Annotation Results.** Visual detail verification exhibited a high level of agreement, with an initial inter-annotator agreement rate of approximately 0.90, reflecting the objective nature of the task. In contrast, context alignment showed lower agreement, with an initial rate of around 0.70, due to its inherently more subjective nature. Instances were initially retained if they received three `Yes` votes for both checks.

**Conflict Resolution and Filtering.**

- Instances that received unanimous `No` judgments from all annotators in either check were directly discarded.

- For cases with conflicting judgments (e.g., one `No`, two `Yes` or any instance with at least one `Not Sure`), a second round of annotation was conducted. During this phase, annotators collaboratively revisited the flagged cases, discussed discrepancies, and reached a consensus decision to ensure consistent quality standards.

- If, after discussion, the final decision still resulted in a `No` for either the visual detail correctness or context plausibility, the instance was removed.

**Filtering Statistics and Error Examples:** Across the dataset, 492 instances were filtered by automatic checks, 371 by manual annotation, resulting in the final sample size of 1093. Common errors detected included hallucinated objects or implausible contexts, further emphasizing the necessity of both automated and human oversight to ensure dataset validity.

## D   Detailed LogiCAM Reasoning Process

Below, we present the step-by-step reasoning workflow of LogiCAM.

**Step 1: Initial Premise Selection.** Given a context set $\mathcal{R}_{\text{real}}$, an image $I$, and access to a VLM, we prompt the model to initiate the reasoning process by selecting relevant information $I_{\text{relevant}} \subseteq \mathcal{C} \cup \mathcal{V}$. The VLM is instructed to prioritize selecting a pair $(\phi, \psi)$ such that a formal inference rule (e.g., Modus Ponens) can be applied. If no such pair exists, the model selects the information it judges most critical for solving the task.

**Step 2: Identify Reasoning Type.** For each selected pair $I_{\text{relevant}}$, we determine the type of reasoning. Symbolic reasoning is applied if the $I_{\text{relevant}}$ contain a pair $(\phi, \psi)$ such that a formal inference rule (e.g., Modus Ponens) can be applied, i.e., $\phi \wedge (\phi \to \chi) \vdash \chi$. Otherwise, commonsense reasoning is used.

**Step 3: Perform Reasoning.** Depending on the reasoning type identified in the previous step, the VLM performs inference to derive new knowledge $K$. For symbolic reasoning, the system applies *syllogistic inference*, a form of deductive reasoning. Specifically, given two selected premises $I_{\text{relevant}} = \{\phi, \psi\}$, the VLM applies formal logical rules to derive a conclusion. For commonsense reasoning, the VLM generates a semantically and contextually plausible implication $\chi$, such that $(I_{\text{relevant}} \to \chi)$, grounded in real-world commonsense knowledge using a VLM. The result of either reasoning process is recorded as $K$.

**Step 4: Check for Completion.** We evaluate whether the current knowledge $K$ is sufficient to determine an answer to the given question. For truth evaluation (True/False/Unknown) questions involving a single hypothesis $H$, if $K \models H$ or $K \models \neg H$, the process terminates with the corresponding label (`True` or `False`); otherwise, it continues. For multiple-choice questions with candidate hypotheses $\{H_1, H_2, H_3, H_4\}$, we apply the reasoning process to each $H_i$ individually and select the one for which $K \models H_i$ holds, if exactly one such $H_i$ exists. If no hypothesis is entailed, or more than one is, we continue the reasoning process. In all cases, the set of relevant information is updated as $I_{\text{relevant}} \leftarrow I_{\text{relevant}} \cup K$, and the procedure is repeated from Step 1. The reasoning loop is bounded by a predefined number of maximum iterations. If no conclusive answer is reached within this limit, the final output is labeled as `Unknown` for truth evaluation questions, or deemed incorrect for multiple-choice questions.

# E   Additional Experiments

## E.1   Using Symbolic Prover on MuSLR

Most existing LLM+solver approaches (e.g., Logic-LM, Logic-LM++, LINC) are designed for text-only reasoning tasks and cannot directly process visual inputs. Extending them to multimodal settings typically requires a vision-language model (VLM), such as GPT-4.1, to translate images into textual descriptions. However, this translation often omits subtle or hard-to-verbalize visual cues.

To illustrate this limitation, we adapted a representative LLM+solver method, Logic-LM [22], by pairing it with a VLM (GPT-4.1) to convert images into text, and compared its performance against LogiCAM on propositional logic (PL) and first-order logic (FOL). (Logic-LM does not support natural language with modalities, NM.) The results are summarized below:

| Model | PL (%) | FOL (%) |
|---|---|---|
| Logic-LM + VLM | 35.14 | 32.65 |
| LogiCAM | 60.44 | 42.55 |

Table 2: Performance comparison of Logic-LM with VLM versus LogiCAM on MuSLR.

These findings demonstrate that simply translating visual information into text is insufficient for effective symbolic reasoning. LogiCAM, which is natively built on VLMs, achieves significantly higher performance since it can directly access visual content. Nonetheless, LLM+solver approaches remain important, and we propose exploring more integrated multimodal LLM+solver frameworks as promising directions for future work.

# F   Atomic Symbolic Logic

Below, we present the atomic symbolic rules used to construct MuSLR.

**Propositional and First-order Logic**

- **Modus Ponens (MP)**

  Propositional:
  $$\big((p \rightarrow q) \wedge p\big) \vdash q$$

  First-order:
  $$\big((\forall x\, (p(x) \rightarrow q(x))) \wedge p(a)\big) \vdash q(a)$$

  If "$p$ implies $q$" and $p$ holds, we may conclude $q$.

- **Modus Tollens (MT)**

  Propositional:
  $$\big((p \rightarrow q) \wedge \neg q\big) \vdash \neg p$$

  First-order:
  $$\big((\forall x\, (p(x) \rightarrow q(x))) \wedge \neg q(a)\big) \vdash \neg p(a)$$

  From $p \rightarrow q$ and $\neg q$ infer $\neg p$.

- **Hypothetical Syllogism (HS)**

  Propositional:
  $$\big((p \rightarrow q) \wedge (q \rightarrow r)\big) \vdash (p \rightarrow r)$$

  First-order:
  $$\big(\forall x\, ((p(x) \rightarrow q(x)) \wedge (q(x) \rightarrow r(x)))\big) \vdash (p(a) \rightarrow r(a))$$

  Chaining two implications into one.

- **Disjunctive Syllogism (DS)**

  Propositional:
  $$\big((p \vee q) \wedge \neg p\big) \vdash q$$

  First-order:
  $$\big((\forall x\, (p(x) \vee q(x))) \wedge \neg p(a)\big) \vdash q(a)$$

  Eliminate a disjunct once the other is shown false.

- **Constructive Dilemma (CD)**
  Propositional:
  $$\big((p \to q) \land (r \to s) \land (p \lor r)\big) \;\vdash\; (q \lor s)$$
  First-order:
  $$\big((\forall x\,((p(x) \to q(x)) \land (r(x) \to s(x)))) \land (p(a) \lor r(a)))\big) \;\vdash\; (q(a) \lor s(a))$$
  From two conditionals and a choice of antecedents, infer a choice of consequents.
- **Destructive Dilemma (DD)** Propositional:
  $$\big((p \to q) \land (r \to s) \land (\neg q \lor \neg s)\big) \;\vdash\; (\neg p \lor \neg r)$$
  First-order:
  $$\big((\forall x\,((p(x) \to q(x)) \land (r(x) \to s(x)))) \land (\neg q(a) \lor \neg s(a)))\big) \;\vdash\; (\neg p(a) \lor \neg r(a))$$
  The "dual" of the constructive dilemma.
- **Biconditional Dilemma (BD)**
  Propositional:
  $$\big((p \to q) \land (r \to s) \land (p \lor \neg s)\big) \;\vdash\; (q \lor \neg r)$$
  First-order:
  $$\big((\forall x\,((p(x) \to q(x)) \land (r(x) \to s(x)))) \land (p(a) \lor \neg s(a)))\big) \;\vdash\; (q(a) \lor \neg r(a))$$
  A mix of constructive and destructive patterns.
- **Commutativity of $\lor$ (CT)**
  Propositional:
  $$(p \lor q) \;\dashv\vdash\; (q \lor p)$$
  First-order:
  $$\forall x\,(p(x) \lor q(x)) \;\dashv\vdash\; \forall x\,(q(x) \lor p(x))$$
  Order of a disjunction doesn't matter.
- **De Morgan's Transformation (DMT)**
  Propositional:
  $$\neg(p \land q) \;\dashv\vdash\; (\neg p \lor \neg q)$$
  First-order:
  $$\neg \forall x\,(p(x) \land q(x)) \;\dashv\vdash\; \exists x\,(\neg p(x) \lor \neg q(x))$$
  Pushing negation inside a conjunction (or quantifier).
- **Conjunction of Conclusions (CO)**
  Propositional:
  $$\big((p \to q) \land (p \to r)\big) \;\vdash\; \big(p \to (q \land r)\big)$$
  First-order:
  $$\forall x\big((p(x) \to q(x)) \land (p(x) \to r(x))\big) \;\vdash\; \forall x\,(p(x) \to (q(x) \land r(x)))$$
  From two implications with the same antecedent, fuse their consequents.
- **Implication Conjunction (IM)**
  Propositional:
  $$(p \to (q \to r)) \;\dashv\vdash\; ((p \land q) \to r)$$
  First-order:
  $$\forall x\,(p(x) \to (q(x) \to r(x))) \;\dashv\vdash\; \forall x\,((p(x) \land q(x)) \to r(x))$$
  Currying/un-currying of implication.
- **Material Implication (MI)**
  Propositional:
  $$(p \to q) \;\dashv\vdash\; (\neg p \lor q)$$
  (No direct first-order analogue listed.)
- **Existential Generalization (EG)** First-order only:
  $$p(a) \;\vdash\; \exists x\,p(x)$$
  From a particular instance infer an existential claim.
- **Universal Instantiation (UI)** First-order only:
  $$\forall x\,p(x) \;\vdash\; p(a)$$
  From a universally quantified claim infer it for an arbitrary constant.

**Extended Multi-variable FOL Rules**

- **MV1**

$$\forall x \forall y\,((p(x) \wedge q(x)) \to r(x,y)) \;\wedge\; \exists u \exists v\,(p(u) \wedge \neg r(u,v)) \;\vdash\; \exists y\,\neg q(y)$$

  If every $p \wedge q$ yields $r$, but there is an instance of $p$ where $r$ fails, then that instance must lack $q$.

- **MV2**

$$\forall x \forall y\,((p(x) \wedge q(x)) \to \neg s(x,y)) \;\wedge\; \forall z\,(r(z) \to p(z)) \;\wedge\; r(a) \wedge s(a,b) \;\vdash\; \neg q(b)$$

  Combines two universally quantified conditionals and a counter-example to force $\neg q(b)$.

- **MV3**

$$\forall x\,\exists y\,\big(p(x) \to q(x,y)\big) \wedge \forall u \forall v\,\big((q(u,v) \wedge r(u,v)) \to s(v)\big) \wedge \exists z \exists k\,(p(z) \wedge r(z,k)) \;\vdash\; \exists w\,s(w)$$

  Chaining an existential-conditional, a universal rule, and an example to derive an existential.

- **MV4**

$$\forall x \forall y \forall z\,(p(x,y,z) \to (q(x,z) \vee r(y))) \;\wedge\; \exists u \exists v \exists w\,(p(u,v,w) \wedge \neg q(u,w)) \;\vdash\; \exists s\,r(s)$$

  If $p$ always gives $q$ or $r$, and for some triple $p$ holds but $q$ fails, then some $r$ must hold.

- **MV5**

$$\forall x\,(p(x) \to \exists y\,r(y,x)) \;\wedge\; p(a) \;\vdash\; \exists z\,r(z,a)$$

  From a universal "$p$ implies an $r$" and one example of $p$, infer the corresponding existential.

- **MV6**

$$\forall x \forall y\,(p(x,y) \vee q(x,y)) \;\wedge\; \exists u \exists v\,\neg q(u,v) \;\vdash\; \exists z \exists w\,p(z,w)$$

  A quantified disjunction plus a counter-example to one disjunct forces the other.

- **MV7**

$$\forall x \forall y\,(p(x,y) \to (q(x) \wedge r(y))) \;\wedge\; p(a,b) \;\vdash\; q(a) \wedge r(b)$$

  From a universal conditional that yields a conjunction, plus an instance, you get both conjuncts.

**Non-monotonic Default-Reasoning Patterns**

- **DRS** (Default Reasoning with Several Defaults) Manages cases where multiple default rules apply at once and may conflict, by finding a consistent combination.

- **DRI** (Default Reasoning with Irrelevant Information) Ensures that adding facts unrelated to a default does not block that default's usual conclusion.

- **DRD** (Default Reasoning with a Disabled Default) Shows how the presence of an exception can "turn off" a default that would otherwise fire.

- **DRO** (Default Reasoning in an Open Domain) Adapts defaults to settings where not all individuals are known or named.

- **REI** (Reasoning about Unknown Expectations I) Allows inferring a default property in the absence of any information to the contrary.

- **REII** (Reasoning about Unknown Expectations II) Refines REI by handling the situation where conflicting expectations might arise.

- **REIII** (Reasoning about Unknown Expectations III) Extends the previous patterns to nested or higher-order expectations.

- **RAP** (Reasoning about Priorities) Introduces a priority ordering among defaults to resolve conflicts in favor of the higher-priority rule.

# G  Ethics Statement

## G.1  Statement

This study adheres to a rigorous ethical framework to ensure the responsible development, evaluation, and deployment of multimodal general-purpose AI models. The key ethical considerations are outlined below. These measures ensure that MuSLR, as a responsible and inclusive framework, continuously contributes to the fair, sustainable, and accountable development of multimodal artificial intelligence.

### G.2 Privacy and Data Protection

The benchmarking and evaluation processes strictly comply with privacy regulations. All tasks and datasets used in MuSLR are carefully curated to exclude any personally identifiable information (PII). Any data obtained from publicly available sources is anonymized and filtered to remove privacy-sensitive content. We are committed to fully adhering to relevant data protection standards, including the General Data Protection Regulation (GDPR) and the California Consumer Privacy Act (CCPA), thereby upholding the highest standards of ethical research practices.

### G.3 Data Collection

All data included in the MuSLR dataset was sourced exclusively from publicly available resources. The data collection protocol is designed to prioritize ethical sourcing, ensuring that contributors' rights are respected, including the right to withdraw their data where applicable. This approach ensures transparency and fairness throughout the dataset construction process.

### G.4 Annotator Compensation

We fully recognize the critical role human annotators play in creating the high-quality MuSLR dataset. All six annotators involved in the project are trained professionals, and they received fair compensation for their work. Annotators were compensated with cash payments upon completion of their assigned tasks. Each annotator was committed to contributing their best efforts to data annotation and quality assurance, ensuring the integrity and reliability of the dataset.

### G.5 Bias and Fairness

We proactively implemented measures to analyze and mitigate potential biases related to gender, ethnicity, language, and other sociocultural factors present in the datasets and evaluation tasks. Our goal is to reduce the risk of perpetuating biases in AI development. While completely eliminating bias remains an ongoing challenge, our commitment to identifying and addressing bias throughout the benchmark development process remains steadfast.

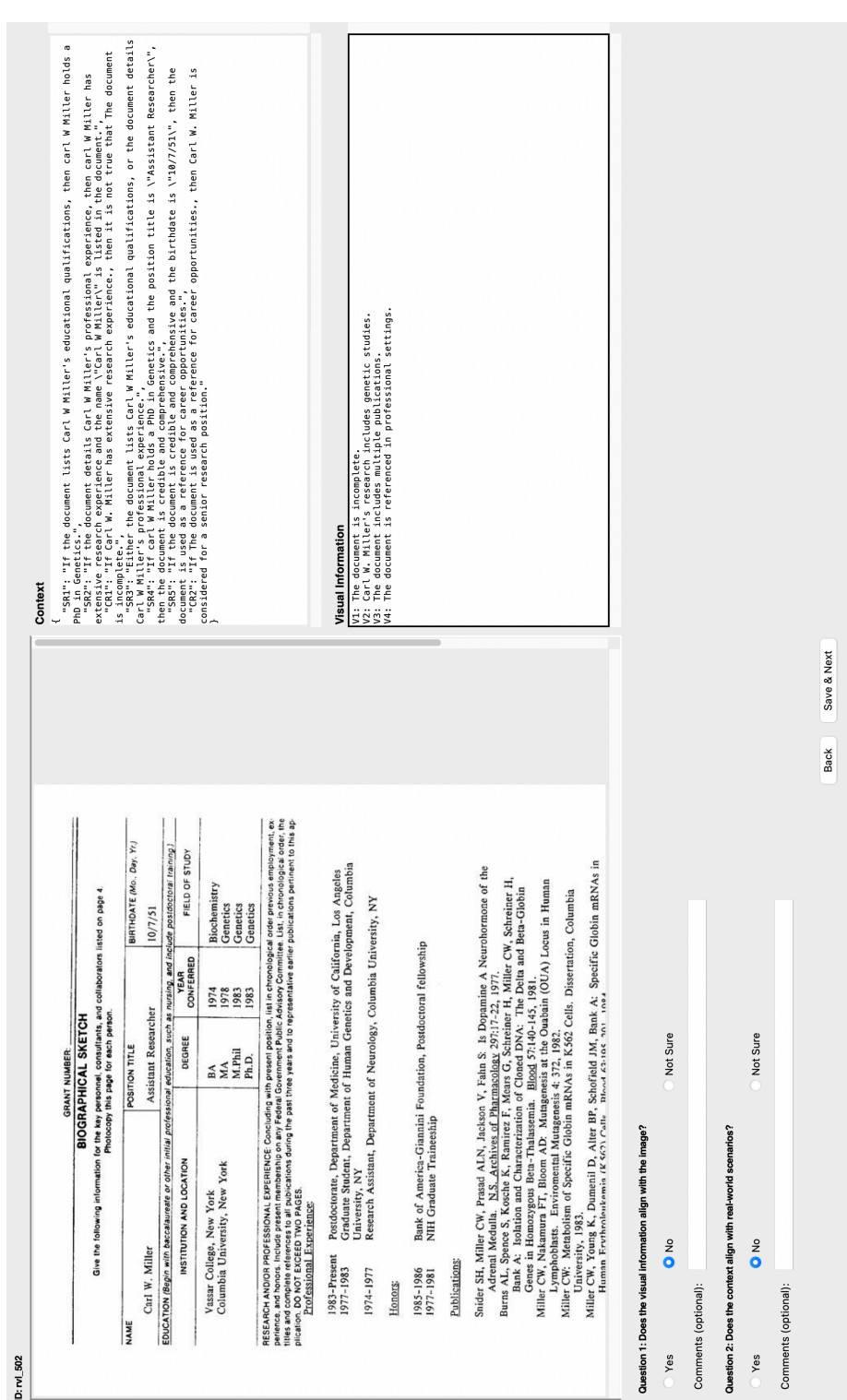

Figure 10: Annotation Interface. We developed a custom interface to streamline the annotation process and reduce annotator effort.

