# OpenReview forum: "MuSLR: Multimodal Symbolic Logical Reasoning"
_NeurIPS.cc/2025/Conference — NeurIPS 2025 poster_

### Official Review · Reviewer_L4xj · 2025-06-09

**Clarity:** 3
**Significance:** 4
**Originality:** 2
**Rating:** 4
**Confidence:** 3

**Summary:**

This paper proposes MuSLR, the first multimodal, multi-domain benchmark for VLMs to perform symbolic reasoning using both textual and visual information. MuSLR requires VLMs to either 1) decide the truth value of an argument based on the given information or 2) select the argument that best corresponds to the input.

In addition, this paper introduces the LogiCAM framework, which is a multi-round CoT procedure that guides in-depth reasoning iteratively. In each step, LogiCAM first attends to the most valuable information at the current step, then decides whether to apply formal symbolic reasoning or commonsense heuristics. Next, it performs reasoning and finally checks if the derived results are sufficient to answer the question.

Empirical evaluations show that MuSLR is a difficult benchmark even for current SOTA VLMs (e.g., GPT-4.1), which achieve an average accuracy of less than 50%. The proposed LogiCAM method significantly boosts model accuracy by around 14.13% on average, demonstrating its effectiveness in iteratively guiding VLMs to solve complex multimodal reasoning problems.

**Questions:**

1.  It is interesting that model performance varies significantly with different families of logic, since they are all used to describe the same scenarios and, therefore, should not produce significantly different results. IMHO this point is not thoroughly discussed in the current manuscript.
    -   **Is the entire dataset within the expressive reach of PL?** Specifically, when constructing the dataset, were there any instances that could not be represented with PL alone, thus requiring the introduction of FOL to capture all necessary information?
        -   If the entire dataset can be expressed within PL, it would be understandable for FOL to compromise model performance, as it introduces many redundant concepts and axioms.
    -   From my understanding, Figure 8 (Error Distribution) is summarized regardless of logic type. **Could the authors provide some quantitative, or at least qualitative, comments on the error distribution for different logic types? From your perspective, why might these differences occur?**
    -   It is worth noting that after applying LogiCAM, the performance divergence between different logic types is greatly reduced. In some cases, FOL even achieves the best results. **Could the authors briefly explain why this happens and what kinds of errors LogiCAM helps to avoid?**
2.  Following up on weakness 1, **is there any way to formally or informally analyze the effectiveness of each module in LogiCAM?** This would make it easier for other researchers to make future improvements and reference the work. I understand that a complete ablation study may not be feasible, so any comments on this would be greatly appreciated.
3.  **What is the relationship between LogiCAM and other CoT frameworks in prior art, for example, for visual or textual reasoning?** Is it a straightforward application of existing methods, or is it tailored for multimodal tasks?

**Ethical Concerns:**

["NO or VERY MINOR ethics concerns only"]

**Final Justification:**

After reading the feedback from the other reviewers, I share some of their concerns regarding the novelty of LogiCAM in the broader context of the field. As I'm not very familiar with the LLM-solver paradigm, I will maintain my current score for now. I remain open to the opinions of the other reviewers on this matter.

**Limitations:**

The proposed MuSLR dataset is valuable; however, the choice of different logic types (PL, FOL, NM) seems somewhat arbitrary and may not be fully informative for downstream VLM evaluation.

**Quality:**

3

**Strengths And Weaknesses:**

**Strengths**

1.  The proposed benchmark is highly relevant to VLM research and could serve as a valuable resource for VLM researchers seeking to improve the general reasoning abilities of VLMs, or for neural-symbolic researchers seeking benchmark tasks for multimodal models.
2.  The proposed LogiCAM approach is intuitive and effective.
3.  The writing is clear, and the paper contains a rich analysis of different aspects of the dataset and the CoT framework, making the paper informative for readers in related research areas.

**Weaknesses**

1.  The design of the proposed LogiCAM framework is largely heuristic. There is no clear ablation study to pinpoint the effectiveness of each module or to determine how it might be improved.
2.  LogiCAM uses VLMs directly to perform reasoning. While this is a natural choice for heuristic commonsense reasoning, it is less reliable for formal symbolic reasoning. This approach leads to the "Incorrect Application of Logical Rules" errors reported in the paper. Given that a symbolic representation of facts arises naturally in the CoT process, the authors might consider incorporating a formal symbolic reasoner to aid this process and provide verifiable results.

---

> ### Author Rebuttal · Authors · 2025-07-29
>
> Thank you for acknowledging the contributions of our work, the effectiveness of our approach, and the clarity of our writing. Please see below for our response to your concerns.
>
> ---
>
> **W1: Ablation study for LogiCAM**
>
> Thank you for raising this important point. We agree that understanding the contribution of each component in LogiCAM is essential for both interpretability and future extensibility.
>
> To address this, we have now conducted an ablation study to evaluate the impact of the three core components in LogiCAM:
>
> | Model Variant         | Accuracy (%) |
> |-----------------------|--------------|
> | LogiCAM               | 61.34        |
> | w/o Symbolic          | 56.20        |
> | w/o Heuristic         | 57.89        |
> | w/o Premise selection | 58.07        |
>
> Key findings from our ablation study show that removing the symbolic module drops performance by over 5 points, omitting heuristic reasoning also hurts results, and disabling premise selection impairs focus and accuracy. These results confirm that LogiCAM’s design, where symbolic reasoning provides rigor, heuristics add flexibility, and premise selection boosts efficiency, works as a coherent, modular system. We will include this analysis in our revision.
>
> ---
>
> **W2: The author might consider incorporating a formal symbolic reasoner**
>
> Thank you for raising this important point. We fully agree that using VLMs directly for formal symbolic reasoning has limitations, given their probabilistic nature and lack of deterministic guarantees.
>
> We did not include these the symbolic reasoner originally because they are designed for text-only tasks and cannot handle visual inputs directly. Adapting them to multimodal settings requires a VLM (e.g., GPT-4.1) to convert images into text, which can omit subtle or hard-to-verbalize visual cues.
>
> To illustrate this, we adapted a typical LLM+solver approach, Logic-LM, using VLM (GPT4.1), to transform images into text and compared it with LogiCAM on PL and FOL (Logic-LM does not support NM):
>
> | Model             | PL (%)  | FOL (%) |
> |------------------|---------|---------|
> | Logic-LM + VLM   | 35.14%  | 32.65%  |
> | LogiCAM          | 60.44%  | 42.55%  |
>
> These results support our view that translating visuals into text is not enough for effective symbolic reasoning. LogiCAM, built on VLMs, performs significantly better since it can directly access the image. However, we agree that LLM+solver approaches are important, and we propose a more integrated multimodal LLM+solver framework as future work.
>
> In the revision, we will 1) Discuss challenges in adapting LLM+solver pipelines to multimodal symbolic reasoning tasks; 2) Cite additional relevant works (e.g., LINC, Logic-LM++, TIC, Divide-and-translate); 3) Expand the related work section (see also our response to Reviewer oQKt, Comment W1).
>
> ---
>
> **Q1.1 Why does model performance vary across logic types despite describing the same scenarios**
>
> We would like to first clarify that the scenarios are not exactly the same. Each instance in the benchmark includes a unique image and text. While a small subset may share similar abstract logical structures, this is not the norm.
>
> Moreover, the expressive power and syntactic complexity of the logic types differ, which can significantly impact model performance, even when the underlying semantics are comparable. For example, PL involves simple atomic statements, making it easier for models to match surface patterns. FOL introduces quantifiers and variable binding, which require structured reasoning and symbol tracking, areas where current VLMs often struggle. NM involves defaults and exceptions, which can sometimes align more closely with the heuristic or commonsense reasoning patterns of pretrained models.
>
> As a result, even with similar scenario content, the reasoning demands and representational challenges vary across logic types, leading to performance differences.
>
> ---
>
> **Q1.2: Is the entire dataset within the expressive reach of PL?**
>
> No, the entire dataset cannot be fully expressed using PL alone. During dataset construction, we encountered many instances where quantification over objects or entities was essential. For example:
>
> ● "If all the people are standing, then the room is full."
>
> ● "There exists a vehicle approaching the crosswalk."
>
> These naturally require first-order constructs, such as universal (∀) or existential (∃) quantifiers and predicate-argument structure, which are not available in PL. Thus, the inclusion of FOL is not redundant. It is required for the semantic coverage of real-world grounded scenarios in the dataset.
>
> We will clarify this point in the revised manuscript and give a few representative examples in the appendix to illustrate cases that necessitate FOL.
>
> ---
>
> **Q1.3: Can you provide error breakdown by logic type?**
>
> Thank you for this suggestion. In the revised version, we will provide error distribution analysis broken down by logic type to complement the aggregate summary in Figure 8. This will include both quantitative differences in error rates and qualitative descriptions of typical failure cases for PL, FOL, and NM.
>
> | Logic | Visual Perception   | Logical Alignment | Heuristic Shortcuts  | Overlook Visual Details | Incorrect Logic | Commonsense Misgen. |
> |-------|------|-------|------|------|---------|---------|
> | PL    | 0.05 | 0.68  | 0.07 | 0.10 | 0.08    | 0.01    |
> | FOL   | 0.05 | 0.44  | 0.11 | 0.16 | 0.17    | 0.06    |
> | NM    | 0.07 | 0.79  | 0.02 | 0.07 | 0.05    | 0.00    |
>
> We have the following observations:
>
> 1. Consistent Alignment Issues Across All Logic Types:
>  A major source of failure across PL, FOL, and NM is logical misalignment between text and image, particularly prominent in NM (79%) and PL (68%). This is consistent with our broader finding that mapping formal logical structures onto multimodal context remains a core challenge for current VLMs.
>
> 2. FOL is the Most Prone to Overlooking and Logical Errors: Overlooking errors are highest in FOL (16%), suggesting the model frequently misses critical details, especially in scenarios requiring reasoning over multiple entities, nested structures, or quantifiers. Incorrect logical rule application is also most common in FOL (17%), which we attribute to its greater symbolic complexity. Unlike PL or NM, FOL involves quantifier binding, variable tracking, and relational reasoning, making it more error-prone for models with less symbolic rigor.
>
> 3. PL Relies Heavily on Symbolic Alignment:
>  While PL avoids many deep logic flaws, its performance heavily depends on correct logical text-image mapping (68% alignment errors). Once aligned, its simpler structure makes it easier for models to apply correct reasoning rules.
>
> 4. NM Reflects High Alignment Difficulty but Few Logical Errors:
>  Despite the highest alignment error rate (79%), NM exhibits the fewest incorrect logic application (5%) or commonsense supplement (0%) errors. This suggests that once alignment is successful, NM reasoning tends to align with the model’s intuitive understanding or default patterns, which may explain its relatively good raw performance.
>
> ---
>
> **Q1.4: Why does LogiCAM reduce the performance gap between logic types?**
>
> LogiCAM narrows the performance gap by imposing a unified, scaffolded reasoning workflow on every problem, regardless of logic type. Instead of allowing the model to rely on its uneven internal heuristics, it breaks each task into the same sequence of steps, identifying premises, selecting rules, instantiating variables, handling exceptions or defaults, and drawing a conclusion. This consistent template forces explicit variable binding in FOL, eliminating scope errors; isolates defaults and overrides in NM, preventing overgeneralization; and mandates multi‑step decomposition even for simple PL tasks, deterring one‑shot pattern matching. By channeling PL, FOL, and NM through the same structured pipeline, LogiCAM standardizes the model’s reasoning process, reducing variance and closing the performance gap between logic types.
>
> We will revise our manuscript to include this analysis.
>
> ---
>
> **Q2: Ablation Study**
>
> Please see details in W1.
>
> ---
>
> **Q3: Is LogiCAM a straightforward application or tailorder for multimodal tasks?**
>
> LogiCAM builds on Chain‑of‑Thought prompting but is specially adapted for multimodal symbolic reasoning. It grounds each step in both visual and textual context, integrates formal inference rules (e.g., Modus Ponens, exception handling) alongside heuristics, and uses logic‑aware prompt templates to enforce structured rule application. These adaptations go well beyond standard CoT methods.
>
> ---
>
> **Response to Limitations**
>
> First, we would like to clarify that the choices of PL, FOL, and NM are in line with text-only symbolic reasoning benchmarks such as LogicBench [1] and Multi‑LogiEval [2] to ensure consistency and comparability. While other logic types like Higher‑Order Logic (HOL) offer greater expressiveness, they bring representational, computational, and annotation challenges beyond current VLM capabilities. Focusing on PL, FOL, and NM lets us deliver a controlled, interpretable benchmark; we plan to explore HOL in future iterations once baseline performance is better understood. We will clarify this rationale in our revision.
>
> [1] LogicBench: Towards Systematic Evaluation of Logical Reasoning Ability of Large Language Models. ACL 2024.
>
> [2] Multi-LogiEval: Towards Evaluating Multi-Step Logical Reasoning Ability of Large Language Models. EMNLP 2024.
>
> ---
>
> Thank you again, and please don't hesitate to reach out if you have any further questions.

---

> > ### Comment · Reviewer_L4xj · 2025-08-06
> >
> > I would like to thank authors for their detailed reply. I hope these information proves useful to future readers.
> >
> > After reading the feedback from the other reviewers, I share some of their concerns regarding the novelty of LogiCAM in the broader context of the field. As I'm not very familiar with the LLM-solver paradigm, I will maintain my current score for now. I remain open to the opinions of the other reviewers on this matter.

---

> ### Author Response · Authors · 2025-08-06
>
> Thank you so much for your thoughtful response. We will include these analyses in our revised version.
>
> We would like to offer a few clarifications regarding the novelty and comparisons with LLM+Solver:
>
> ---
>
> First, regarding the novelty of LogiCAM:
> The core contribution of LogiCAM lies in its **integration of symbolic reasoning with flexible, heuristic-like strategies in multimodal settings, enabling it to tackle a broader range of real-world problems**. To the best of our knowledge, no prior work combines these elements in this way. **Existing approaches typically focus either on purely heuristic-based reasoning or purely symbolic methods, overlooking the fact that real-world problem-solving often involves interleaving both types of reasoning**. This point is further supported by our ablation study (see response to W1), which demonstrates the effectiveness of this combination.
>
>
> While Reviewer PLuU referenced Proof-of-Thought, we highlight two key differences:
> (i) Proof-of-Thought focuses **exclusively on symbolic reasoning, lacking the flexible heuristic component**, which limits its applicability in less structured or ambiguous scenarios;
> (ii) its implementation **has not been open-sourced**, making reproducibility and benchmarking difficult. In contrast, LogiCAM is fully open-sourced, promoting both transparency and accessibility.
>
> ---
>
> Second, on LLM+Solver Comparisons: We would like to clarify that **Solver approaches can only process textual input and cannot directly access visual information**. Therefore, running such a paradigm on MuSLR is not fully representative. That said, we still conducted a variant where a VLM is used to transform the image into text, which is then passed to a Solver.
>
> | Model              | PL (%) | FOL (%) |
> |--------------------|--------|---------|
> | Logic-LM + VLM     | 35.14% | 32.65%  |
> | LogiCAM            | 60.44% | 42.55%  |
>
> **The results show significantly lower performance compared to LogiCAM, highlighting that directly adapting a solver to a multimodal setting is not effective.** However, we consider this a valuable direction for future work and will include a discussion of it in our revised manuscript.
>
> ---
>
> We appreciate the reviewer’s openness to continued discussion, and we hope these clarifications help contextualize our contributions more clearly.

---

> ### Author Response · Authors · 2025-08-07
>
> Dear Reviewer L4xj,
>
> I would like to provide an update regarding Reviewer PLuU’s concern about the LLM+solver paradigm, which appears to be similar to the concern you expressed. We have addressed Reviewer PLuU’s feedback, which led to a recommendation to accept. We have also committed to updating the revised version accordingly.
>
> You may refer to our discussion with Reviewer PLuU for further details and to see if the response also addresses any concerns you might have. If you have any questions or would like to discuss further, please don’t hesitate to reach out. We’d be happy to continue the conversation.

---

### Official Review · Reviewer_CiCb · 2025-07-01

**Clarity:** 4
**Significance:** 3
**Originality:** 3
**Rating:** 5
**Confidence:** 4

**Summary:**

This paper introduces Multimodal Symbolic Logical Reasoning (MuSLR), a novel task requiring Vision Language Models (VLMs) to perform formal logical reasoning by integrating visual and textual inputs. The authors create MuSLR-Bench, a benchmark dataset with 1,093 instances across 7 domains featuring reasoning depths from 2-9 steps, and propose LogiCAM, a modular framework that improves GPT-4.1's performance by 14.13%.

**Questions:**

1. Is it a typo in line 24 for the word "text"? Should it be "visual" or anything else?
2. Seems that all the rules are first order logic rules right? Did you consider a higher-order case to represent a more natural real-world case?

**Ethical Concerns:**

["NO or VERY MINOR ethics concerns only"]

**Final Justification:**

The authors have provided further explanations of their experiment results. Although they didn't include higher-order logic into their benchmark, i think this work is good as an pioneer. I confirm my rating.

**Quality:**

4

**Strengths And Weaknesses:**

Strenghts: I think this paper is really great and deserves a publication. The proposed benchmark addresses a key challenge that most previous works just focused on symbolic reasoning in a single text modality. The proposed method is also reasonable to me. Furthermore, the code is published for reproducibility.

Weaknesses: Although authors have done a lot of experiments, a deeper analysis of the results will better help the community.

---

> ### Author Rebuttal · Authors · 2025-07-29
>
> Thank you very much for your positive feedback and support. We're glad you find the benchmark meaningful and the method and reproducibility valuable. We will address your concerns below.
>
> ---
>
> **W1: Deeper analysis of the results**
>
> Regarding your suggestion for deeper analysis, we plan to include additional analysis  in the revision to better support the community, including:
>
> 1. Error breakdown by logic type (e.g., PL, FOL, NM), rather than by method alone, to better understand where models struggle across different logical structures (see Q1.3 response to reviewer L4xj in detail).
>
> 2. Deeper analysis on why models perform differently on different logic types (see W3 response to reviewer PLuU in detail).
>
> 3. Ablation study on LogiCAM with both quantitative and qualitative analysis to pinpoint the effectiveness of why each modular design works in the proposed method (see W1 response to reviewer L4xj in detail).
>
> 4. Comparison with the LLM+solver paradigm with both quantitative and qualitative analysis to explain the limitations of LLM+solver in the MuSLR task and what we can do in the future (see W1 response to reviewer PLuU in detail).
>
> We appreciate your constructive input and will ensure the revised version incorporates these enhancements.
>
> ---
>
> **Q1: Regarding line 24 – possible typo for "text"**
>
> This is not a typo. Our intent was to emphasize that most existing symbolic reasoning benchmarks are designed for a single modality, text, without integrating visual information. However, we agree that the current phrasing may be unclear, and we will revise that sentence in the final version to more explicitly convey this point.
>
> ---
>
> **Q2: Are all the rules first-order logic? Have you considered higher-order logic?**
>
> While many of the rules used are from first-order logic (FOL), we also include reasoning patterns from propositional logic (PL) and non-monotonic logic (NM). Our choice of these three categories follows the precedent set by existing textual symbolic reasoning benchmarks (e.g., LogicBench [1], Multi-LogiEval [2]), and we extend them to the multimodal setting for the first time.
>
> Regarding higher-order logic (HOL), we agree it can represent more expressive and natural reasoning structures. However, we chose not to include higher-order logic in this initial benchmark for several reasons. First, HOL introduces significantly higher representational and computational complexity, which may overwhelm current vision-language models and make systematic benchmarking difficult. Second, grounding higher-order reasoning in real-world visual scenarios is much more ambiguous and difficult to verify, both for humans and machines. Besides, our goal is to provide a controlled, interpretable benchmark that can facilitate meaningful comparisons across methods. Including HOL would introduce another layer of abstraction that could obscure insights at this early stage.
>
> That said, we see higher-order multimodal reasoning as a valuable future direction. A promising extension would involve structured tasks requiring reasoning over sets of entities, predicates over predicates, or temporal/event-based logic. We hope MuSLR can serve as a foundational benchmark to eventually support such higher-order capabilities.
>
> We will clarify these design choices and propose HOL as a future direction in the revised manuscript. Thank you again for highlighting this point.
>
> [1] LogicBench: Towards Systematic Evaluation of Logical Reasoning Ability of Large Language Models. ACL 2024.
>
> [2] Multi-LogiEval: Towards Evaluating Multi-Step Logical Reasoning Ability of Large Language Models. EMNLP 2024.
>
> ---
>
> Thank you again, and please don't hesitate to reach out if you have any further questions.

---

> > ### Comment · Reviewer_CiCb · 2025-08-01
> >
> > Thanks for your answers. I will keep my score.

---

> ### Author Response · Authors · 2025-08-01
>
> We sincerely appreciate your positive feedback and recognition of our work. Thanks again!

---

### Official Review · Reviewer_oQKt · 2025-07-02

**Clarity:** 3
**Significance:** 2
**Originality:** 2
**Rating:** 4
**Confidence:** 3

**Summary:**

This paper studies VLMs on multimodal symbolic logical reasoning tasks. First, the authors propose a new benchmark MuSLR for this task and evaluate multiple state-of-the-art VLMs on MuSLR, finding that they all struggle with multimodal symbolic reasoning. To this end, the authors then propose LogiCAM to improve the performance of existing VLMs on this task.

**Questions:**

What is the differences between the proposed MuSLR task and a task that receives multimodal input and includes a symbolic reasoning process?

**Ethical Concerns:**

["NO or VERY MINOR ethics concerns only"]

**Final Justification:**

The contribution of this work is solid, featuring a carefully designed benchmark and a novel framework to address key challenges in the MuSLR task. However, since similar tasks have been explored in prior literature, this may somewhat diminish the novelty of the paper.

**Limitations:**

yes

**Quality:**

3

**Strengths And Weaknesses:**

Strengths:
1. The contribution of this work is solid. After introducing of the background and motivation, the paper 1) raises a new task MuSLR with a clear definition, 2) introduces MuSLR-Bench as a benchmark for MuSLR task including detailed data statistics and construction process, and 3) proposes LogiCAM as a framework to adress challenging issues in MuSLR task with a clear flowchart and process description. Exhaustive experiments and in-depth analyses offer insights of this task.
2. The writing of this work is clear and easy to follow.
3. Part of the code and dataset is provided.

Weaknesses:
1. The MuSLR seems not entirely new. Some previous work like [1] also uses multimodal input and includes a symbolic reasoning process. The authors should better discuss about their differences with such prior studies.

[1] Symbol-LLM: Leverage Language Models for Symbolic System in Visual Human Activity Reasoning

---

> ### Author Rebuttal · Authors · 2025-07-29
>
> Thank you for acknowledging the contributions of our work, including the exhaustive experiments and in-depth analyses. We would like to address your questions in detail below.
>
> ---
>
> **W1: Differences with prior studies**
>
> Thank you for highlighting the relevance of prior work such as Symbol-LLM. We appreciate the opportunity to clarify the novelty of MuSLR and how it differs from existing studies. While Symbol-LLM and similar works incorporate multimodal input and symbolic reasoning, MuSLR introduces a benchmark, not a specific method, focused on systematically evaluating multistep symbolic reasoning across grounded multimodal contexts. Unlike Symbol-LLM, which centers on visual activity reasoning, MuSLR provides a task suite covering diverse logic types (PL, FOL, NM) and varying reasoning depths.
>
> MuSLR also formalizes reasoning through explicit logical structures and rule compositions (e.g., Modus Ponens, Syllogism, Non-Monotonic Reasoning), each grounded in real-world contexts. In contrast, Symbol-LLM emphasizes scene understanding without modeling or benchmarking formal logic chains.
>
> Another key distinction is MuSLR’s hybrid reasoning design, combining formal logic with commonsense heuristics to handle ambiguity, something not explored in Symbol-LLM. Finally, MuSLR offers a broader range of logic types and controlled complexity to study model limitations in a structured way.
>
> We will revise the related work section to include Symbol-LLM and clarify these differences in scope, formulation, and goals (as also noted in our response to Reviewer PLuU, Comment W3). Thank you again for drawing this connection.
>
> ---
>
> **Q1: Differences between MuSLR and other symbolic reasoning tasks**
>
> We would like to clarify the distinction between MuSLR and general tasks that involve multimodal input with symbolic reasoning:
>
> 1. Narrowed Focus on Formal Logical Reasoning: "Symbolic reasoning" is indeed a broad term, encompassing domains such as mathematical problem solving, program synthesis, and abstract planning. While many existing multimodal tasks include symbolic elements in a loose sense, MuSLR explicitly focuses on formal logical reasoning, grounded in classical logic types such as propositional logic (PL), first-order logic (FOL), and non-monotonic logic (NM). This aspect is largely underexplored in prior multimodal benchmarks, which tend to focus on either mathematical or implicit logical reasoning without a clearly defined logical structure.
>
> 2. Hybrid Symbolic + Heuristic Reasoning Paradigm: Another key difference is that MuSLR explicitly requires a combination of symbolic and heuristic reasoning. This reflects human cognitive behavior: we use formal logic when precise conclusions are needed, and rely on heuristics or intuitive reasoning when information is ambiguous or incomplete. MuSLR is designed to evaluate this hybrid reasoning ability by incorporating tasks that cannot be solved purely by logic or intuition alone, something that existing benchmarks rarely capture systematically.
>
> Together, these aspects make MuSLR not just another multimodal symbolic task, but a structured benchmark for studying how vision-language models handle formal, compositional, and flexible reasoning in grounded, real-world contexts.
> We will revise the manuscript to better articulate these differences and highlight the novelty of MuSLR’s focus and design.
>
> ---
>
> Thank you again, and please don't hesitate to reach out if you have any further questions.

---

> > ### Comment · Reviewer_oQKt · 2025-08-06
> >
> > I would like to thank the authors for the detailed response. I will maintain my score.

---

> ### Author Response · Authors · 2025-08-06
>
> Thank you once again for recognizing the contribution of our work!

---

### Official Review · Reviewer_PLuU · 2025-07-02

**Clarity:** 2
**Significance:** 4
**Originality:** 3
**Rating:** 5
**Confidence:** 5

**Summary:**

This paper introduces a task requiring models to perform formal logical reasoning over combined visual and textual inputs. The authors create (1) MuSLR-Bench, a dataset of ~1k instances spanning 7 domains with ground-truth reasoning chains using propositional logic, first-order logic, and non-monotonic logic. They (2) evaluate 7 VLMs, finding that GPT-4.1 achieves 46.8% accuracy. They (3) propose LogiCAM, a prompt-driven framework that contains premise selection, reasoning type identification, and formal inference components, which improves GPT-4.1's performance by 14%.

**Questions:**

1. Dataset composition says 35+976 ≠ 1,093. May you please elaborate on the composition?
2. Are you using a solver?
3. Figure 7 shows accuracy performance degradation at higher depths. Why?
4. Could you please elaborate on Weakness 3.
5. I am willing to revise my scores upwards if you can address weakness 1.
6. Could you please elaborate on your dataset construction methodology. I find this very interesting. More specifically, I want to understand how you deal with the following
	1. How do you do semantic disambiguation in the autoformalization process.
	2. Is this quantifier free? How do you deal with quantifiers, which may add significant complications to autoformalization.
	3. How do you formalize inherently imprecise natural language statements? How do you deal with implicit meaning in language queries? Did you face any of these issues
	4. How do you go about determining the necessary entities, properties, and relations for the problem's domain, defining the universe of discourse. How do you go about choosing the appropriate level of abstraction? How do you frame invariants?

**Ethical Concerns:**

["NO or VERY MINOR ethics concerns only"]

**Final Justification:**

The authors have benchmarked against LogicLM, and will clarify in their final manuscript that their LogiCAM technique is not using a solver, does not provide guarantees, and is a strong prompting based baseline. My concerns are resolved.

**Limitations:**

Yes

**Quality:**

2

**Strengths And Weaknesses:**

**Strengths**
1. (Originality and Significance) The task genuinely important, and the dataset contribution is timely. As mentioned by the authors, prior work like LogicVista and VisuLogic don't test multimodal formal logical rules, but just do logical reasoning in visual contexts. The novelty here comes from the explicitness of symbolic rule annotation in the dataset (compared to implicit logical reasoning in many VQA datasets).  The benchmark will likely be valuable for the community.
2. (Quality) The experimental design is sound, with multiple evaluation metrics. There are some actionable insights, particularly the finding that 70% of failures stem from cross-modal alignment issues.
3. (Clarity) The paper contains helpful visualizations, and workflow diagrams to communicate the approach.

**Weaknesses**
1. (Quality) I think the paper has a glaring gap in comparing LogiCAM against techniques presented in the literature. LLMs+Solver paradigms already exist. For example : Logic-LM (cited) and many more papers (not cited) ; already combines LLMs with symbolic solvers, decomposing reasoning into symbolic steps. Multimodal symbolic reasoning has also been shown to work in papers like Proof-of-Thought (not cited). If you have a new framework to replace these, you should compare it against adapted versions of other frameworks previously proposed, and not just the base language models.
2. The distinction between "symbolic" and "heuristic" reasoning in LogiCAM seems arbitrary and inadequately defined.
3. Why certain logic types (FOL vs PL vs NM) have different difficulty levels needs better understanding. Lines 233-240 do not line up with standard understanding of complexity. In non-monotonic reasoning, you may invalidate previous conclusions, and the solver would need to handle belief revision and consistency maintenance, which would make it more complex than first order logic.
4. There are a number of typos strewn across the paper, which could use thorough proof-reading. Here are some examples:
	1. Fig 1 : Step 4 "pposing" instead of "opposing"; Step 3: Minor Premise 1: "woad" instead of "road"
	2. Some references use "CoRR, abs/XXXX" while others use full venue names
	3. Table 1 Formatting InstructBlip row: "33,33" should be "33.33" (comma instead of period)
	4. Figure 9 : Context text says "arounod", should be "around".

---

> ### Author Rebuttal · Authors · 2025-07-29
>
> Thanks to the reviewer for acknowledging the significance, originality and quality of the work. We would like to address your concerns and questions in detail below.
>
> ---
>
> **W1: Comparing LogiCAM against techniques presented in the literature**
>
> We appreciate the reviewer’s suggestion to compare LogiCAM with LLM+solver methods like Logic-LM. We will revise the manuscript to better acknowledge this line of work and expand the related work section.
>
> We did not include these methods originally because most LLM+solver approaches (e.g., Logic-LM, Logic-LM++, LINC) are designed for text-only tasks and cannot handle visual inputs directly. Adapting them to multimodal settings requires a VLM (e.g., GPT-4.1) to convert images into text, which can omit subtle or hard-to-verbalize visual cues.
>
> To illustrate this, we adapted a typical LLM+solver approach, Logic-LM, using VLM (GPT4.1), to transform images into text and compared it with LogiCAM on PL and FOL (Logic-LM does not support NM):
>
> | Model             | PL (%)  | FOL (%) |
> |------------------|---------|---------|
> | Logic-LM + VLM   | 35.14%  | 32.65%  |
> | LogiCAM          | 60.44%  | 42.55%  |
>
> These results support our view that translating visuals into text is not enough for effective symbolic reasoning. LogiCAM, built on VLMs, performs significantly better since it can directly access the image. However, we agree that LLM+solver approaches are important, and we propose a more integrated multimodal LLM+solver framework as future work.
>
> In the revision, we will 1) Discuss challenges in adapting LLM+solver pipelines to multimodal symbolic reasoning tasks; 2) Cite additional relevant works (e.g., LINC, Logic-LM++, TIC, Divide-and-translate; happy to include others if you believe there are important and relative methods we may have missed.); 3) Expand the related work section (see also our response to Reviewer oQKt, Comment W1).
>
> Regarding **Proof-of-Thought**, we acknowledge its relevance to multimodal symbolic reasoning and will include and cite it in our revised manuscript. However, due to the lack of a public codebase, we were unable to replicate its results within the rebuttal timeframe.
>
> ---
>
> **W2. Distinction between "symbolic" and "heuristic"**
>
> We thank the reviewer for raising this important point. To address this, we would like to point out that a formal definition of these two types of reasoning is already provided in the Appendix (Lines 583–594) of our manuscript. In brief:
>
> ● Symbolic reasoning follows formal logic, using rules like modus ponens to derive conclusions from explicit premises.
>
> ● Heuristic reasoning, by contrast, encompasses informal, intuitive, or commonsense reasoning strategies. These are employed in situations where formal logical structure is unavailable or insufficient, such as dealing with ambiguous visual information, implicit information, or assumptions.
>
> This distinction reflects our goal of combining structured symbolic inference with the flexible reasoning capabilities of VLMs, as not all information can be explicitly encoded in formal logic in real-world situations. We will clarify this distinction earlier in the main text and better reference the definitions from the appendix in the revision.
>
> ---
>
> **W3. Why certain logic types (FOL vs PL vs NM) have different difficulty levels needs better understanding**
>
> Thank you for highlighting this. Our analysis focuses on empirical model performance, not formal complexity. While NM is theoretically more complex than FOL, we observe that VLMs perform better on NM. This reflects model behavior, not a contradiction of theoretical complexity.
>
> We believe this is because NM aligns better with heuristic reasoning, which VLMs handle well, whereas FOL requires precise logic (e.g., quantifiers, variable binding), which remains challenging for VLMs due to its probabilistic nature.
>
> This trend aligns with existing work like Multi-LogiEval [2], which shows that in complex scenarios (e.g., depth >= 5), NM shows higher accuracy than PL and FOL due to the richer rule combinations that enhance contextual reasoning. These results suggest that NM’s empirical advantage stems from how models leverage its structure, not its theoretical complexity.
>
> We appreciate the reviewer’s comment for helping us improve the clarity of this distinction and will add this to the revision.
>
> [2] Multi-LogiEval: Towards Evaluating Multi-Step Logical Reasoning Ability of Large Language Models. EMNLP 2024.
>
> ---
>
> **W4. Typos in the paper**
>
> Thank you so much for your careful reading of our paper. We will thoroughly proofread the entire manuscript and correct all identified issues.
>
> ---
>
> **Q1: Elaborate on the dataset composition**
>
> Thank you for raising this question. To clarify:
>
> ●The 35 atomic rules form the basis of our symbolic reasoning tasks.
>
> ●We then create permutations and combinations of rules manually. (e.g., applying one rule after another) to make sure it is meaningful.
>
> ●However, during dataset construction, we filtered out semantically invalid or redundant combinations (see Appendix C). After this filtering, we retain 976 valid rule combinations.
>
> ●Importantly, the total number of instances in the dataset is not equal to the number of rule combinations, because each rule combination can appear in multiple distinct grounded scenarios. That is, different images and premises may instantiate the same abstract logic pattern, resulting in a total of 1,093 instances in the dataset.
>
> We will revise the dataset section of the paper to clarify the distinction between atomic rules, combinations, and grounded instances, and we thank the reviewer again for catching this ambiguity.
>
> ---
>
> **Q2: Are u using a solver?**
>
> No, LogiCAM is a prompt-guided framework without an external symbolic solver. It serves as a baseline for the MuSLR benchmark to explore how current VLMs handle multimodal symbolic reasoning. Integrating external solvers is a valuable direction, which we will highlight as future work (see W1 for the reason).
>
> ---
>
> **Q3: Figure 7 shows accuracy performance degradation at higher depths. Why?**
>
> As reasoning depth increases, more inference steps are needed, making it harder for the model to stay coherent and avoid errors. This added complexity naturally leads to performance degradation.
>
> ---
>
> **Q4: Elaborate on Weakness 3.**
>
> Please see W3 in detail.
>
> ---
>
> **Q5: I am willing to revise my scores upwards if you can address weakness 1.**
>
> Thank you for your consideration. We appreciate your openness. Please see our response to W1, and let us know if you have any further questions.
>
> ---
>
> **Q6.1: How do you do semantic disambiguation in the autoformalization process.**
>
> First, in Step 3, we align abstract logical rules with grounded instances by using GPT-4o to extract fine-grained visual details and relevant textual context. Second, all grounded rules are manually curated and verified to ensure they preserve the intended semantics of the original logical form; any ambiguous or misaligned cases are flagged and discarded.
>
> ---
>
> **Q6.2: Is this quantifier free? How do you deal with quantifiers, which may add significant complications to autoformalization.**
>
> We do not limit ourselves to quantifier-free logic but acknowledge that quantifiers, especially nested ones, add significant complexity. Our dataset includes simple universal and existential quantifiers (e.g., "All X are Y", "There exists an X such that Y"), balance logical complexity and maintain meaningful, human-interpretable reasoning. Quantifiers are handled through verified templates with substitution schemas, ensuring correct alignment between natural language and formal logic.
>
> ---
>
> **Q6.3: How do you formalize inherently imprecise natural language statements? How do you deal with implicit meaning in language queries? Did you face any of these issues?**
>
> Yes, we encountered this challenge, as imprecise or underspecified language is common in real-world contexts. That is why we incorporate commonsense/heuristic reasoning alongside formal rules in a hybrid format (see W2), allowing the model to handle ambiguity more effectively. Instances with problematic implicit assumptions are revised or removed through rule-based and manual review (see Quality Control), ensuring soundness in noisy multimodal settings.
>
> ---
>
> **Q6.4: How do you go about determining the necessary entities, properties, and relations for the problem's domain, defining the universe of discourse. How do you go about choosing the appropriate level of abstraction? How do you frame invariants?**
>
> We use a context-driven approach to define the logical structure and abstraction level for each instance, grounded in both visual and textual information. Entities, properties, and relations are extracted using GPT-4o from images and complemented by retrieved text to build a dynamic universe of discourse and verified by trained students afterwards. The abstraction level is chosen to balance logical expressiveness with interpretability, favoring mid-level logic that supports patterns like Modus Tollens or non-monotonic reasoning without requiring deep ontologies or complex quantifiers, as this allows current vision-language models (VLMs) to reason effectively while keeping the logic human-interpretable and tractable for evaluation.
>
> Invariants are implicitly modeled as conditions expected to remain true throughout the reasoning process unless explicitly negated. For example, if a scene establishes that “the road is dry,” this holds across reasoning steps unless a new fact (e.g., “it started raining”) overrides it. We ensure such consistency through expert-designed rules and manual verification.
>
> This approach ensures reasoning instances are logically sound, semantically grounded, and aligned with real-world multimodal contexts.
>
> ---
>
> Thank you again, and please don't hesitate to reach out if you have any further questions.

---

> ### Comment · Reviewer_PLuU · 2025-08-07
>
> Thank you for the detailed rebuttal and for your willingness to engage with the feedback. The paper is stronger with the addition of the LogiLM. Good work. To avoid confusion, I recommend framing the difficulty of logic types as explicitly described in the rebuttal, in the revised manuscript.
>
> The core of my primary objection is as follows : The paper's introduction effectively argues for the need for rigorous, precise, and verifiable reasoning, especially in high-stakes scenarios. This is the key promise of symbolic logic. However, the proposed LogiCAM framework replaces the verifiable, deterministic application of logical rules with a prompt-guided, probabilistic LLM generation step. While you define a distinction between "symbolic" and "heuristic" reasoning in the appendix, in practice, both are executed by the same black-box LLM. The system is prompted to apply Modus Ponens, but there is no external mechanism to guarantee it has done so correctly, or to verify that it hasn't arrived at a plausible-sounding conclusion through a heuristic shortcut. The very benefits of symbolic reasoning i.e. its verifiability and soundness, are lost when the "solver" is an LLM that is merely emulating the process. This introduces the potential for the exact kind of subtle logical errors that formal systems are meant to prevent.
>
> The MuSLR benchmark is an important and timely contribution to the field. The paper correctly identifies a gap in evaluating formal, multimodal reasoning. However, I maintain that there is a significant disconnect between the motivation for and the claims about LogiCAM. It is an advanced and effective Chain-of-Thought prompting framework that improves a VLM's ability to follow logical structures, but it is not a symbolic reasoning system in the formal sense. The framework's reliance on LLM-based inference for every step undermines the claims of "rigorous," "formal," and "verifiable" deduction.
>
> I appreciate the Logic-LM reproduction, but because of this fundamental gap between the paper's claims and the method's implementation, I cannot improve the rating beyond weak reject for now.
>
> I would be open to reconsidering a higher score if the paper were reframed to present LogiCAM as a state-of-the-art method for approximating symbolic reasoning with VLMs, rather than a framework that performs formal logical inference.

---

> ### Author Response · Authors · 2025-08-07
>
> Thank you very much for your thoughtful and constructive feedback. It has provided valuable insights!
>
> Firstly, we greatly appreciate your acknowledgment of the inclusion of Logic-LM comparisons and the framing of different logic types. We **will certainly incorporate this framing more explicitly in the revised manuscript**, as you suggested. We’re also grateful for your recognition of the MuSLR benchmark as an important and timely contribution to the field. Your comments affirm the importance of evaluating formal multimodal reasoning and further motivate our efforts in this direction.
>
> Regarding the motivation behind LogiCAM, our intention was to propose an intuitive, accessible, and flexible baseline to encourage future research on MuSLR, rather than to fully solve the task. To that end, we adopted a carefully designed prompting workflow that blends symbolic and heuristic reasoning to approximate structured reasoning within the current capabilities of VLMs. We hope this approach serves as an inspiration for future work. For example, future work could explore how symbolic solvers might be effectively integrated into multimodal settings to provide the formal logical guarantees that LogiCAM currently lacks.
>
> We agree with your important observation: because the symbolic reasoning steps in LogiCAM are executed by a probabilistic model, there is no mechanism of the logical guarantees. Your point about the gap between the motivation for formal reasoning and the implementation of LogiCAM is well taken. **In response, we will revise the framing of LogiCAM in the paper to clarify that it is not a formal symbolic reasoning system, but rather a state-of-the-art method for approximating symbolic reasoning using VLMs.** We will also reposition it as a strong baseline that highlights current limitations and points toward promising directions for future work, such as the integration of external symbolic solvers in multimodal settings.
>
> Finally, we sincerely appreciate your openness to reconsidering a higher evaluation in light of this reframing clarifications. Your feedback has been invaluable in helping us improve the clarity and rigor of the paper. Please don’t hesitate to reach out if you have further questions or suggestions. We’re happy to continue the discussion.

---

> > ### Comment · Reviewer_PLuU · 2025-08-07
> >
> > Thanks, that sounds good to me. I hope you’ll complete the changes you promised. I’ll upgrade my score to accept now, because my concerns have been resolved with these changes.

---

> ### Author Response · Authors · 2025-08-07
>
> Thank you very much for your valuable feedback on the paper, as well as your recognition of its important contribution. We truly appreciate it and will revise the paper accordingly!

---

### Note · Authors · 2025-08-12

We sincerely thank all reviewers for their valuable insights and constructive feedback. **We are encouraged by the uniformly positive evaluations of our work.** We would like to re-emphasize the importance of the work:
- Propose the first multimodal symbolic logical reasoning task grounded in formal logic, vital for high-stakes domains like autonomous driving and medical diagnosis, where reasoning error could cause critical consequences.
- MuSLR benchmark shows that even the best VLMs achieve only 46.8% accuracy.
- LogiCAM improves the best baseline performance by 14.13%
- Provide comprehensive analysis and insights to advance future multimodal reasoning.


**The contributions are recognized by all reviewers**, summarized as follows:
- Important and timely contribution; originality, significance, quality, and clarity (Reviewer PLuU)
- Solid and clear (Reviewer oQKt)
- Great and worth publication (Reviewer CiCb)
- Highly relevant to VLM research; proposed method is intuitive and effective, with clear writing and rich analysis (Reviewer L4xj)

Reviewers raised three main concerns: expanding the discussion of related work (e.g., LLM+solver, Symbol-LLM, and other multimodal symbolic reasoning benchmarks), framing our method as an approximate rather than a formal symbolic solver, and adding experiments and analyses, including ablation studies and error breakdowns by logic type. **In response, we will incorporate the following revisions, all of which have been thoroughly addressed during rebuttal:**
- Add the comparison with Logic-LM and ablation studies, with corresponding analyses already provided in the rebuttal (See discussion with Reviewer PLuU and L4xj).
- Expand the discussion of prior work to better position our contributions within the broader literature (See discussion with Reviewer PLuU and oQKt).
- Include an error breakdown by logic type, along with detailed analysis, as presented in the rebuttal (See discussion with Reviewer PLuU and L4xj).
- Revise the motivation of LogiCAM to clearly emphasize its role as an approximate symbolic solver (See discussion with Reviewer PLuU).

We are grateful for the reviewers’ recognition of our contributions to the community and their helpful suggestions, which will meaningfully improve the quality of our work. We believe this paper will make an important and positive contribution to the community, inspiring further research and fostering progress in this rapidly evolving area.

Kind regards.

---

### Decision · Program_Chairs · 2025-09-17

**Decision:**

Accept (poster)

**Comment:**

This paper introduced an important benchmark for multimodal symbolic logical reasoning. Four reviewers unanimously agree with that this paper made significant contributions and recommend an acceptance. After a careful check, the AC agrees with the reviewers and recommend an acceptance.